# A microscopic Kondo lattice model for the heavy fermion antiferromagnet CeIn₃

W. Simeth [1,2], Z. Wang [3,4,11], E. A. Ghioldi[3], D. M. Fobes [5], A. Podlesnyak [6], N. H. Sung[5], E. D. Bauer [5], J. Lass [7], S. Flury[1,2], J. Vonka [1], D. G. Mazzone [7], C. Niedermayer [7], Yusuke Nomura [8], Ryotaro Arita [8,9], C. D. Batista [3,10], F. Ronning[5] & M. Janoschek [1,2,5] ✉

Electrons at the border of localization generate exotic states of matter across all classes of strongly correlated electron materials and many other quantum materials with emergent functionality. Heavy electron metals are a model example, in which magnetic interactions arise from the opposing limits of localized and itinerant electrons. This remarkable duality is intimately related to the emergence of a plethora of novel quantum matter states such as unconventional superconductivity, electronic-nematic states, hidden order and most recently topological states of matter such as topological Kondo insulators and Kondo semimetals and putative chiral superconductors. The outstanding challenge is that the archetypal Kondo lattice model that captures the underlying electronic dichotomy is notoriously difficult to solve for real materials. Here we show, using the prototypical strongly-correlated anti-ferromagnet CeIn₃, that a multi-orbital periodic Anderson model embedded with input from ab initio bandstructure calculations can be reduced to a simple Kondo-Heisenberg model, which captures the magnetic interactions quantitatively. We validate this tractable Hamiltonian via high-resolution neutron spectroscopy that reproduces accurately the magnetic soft modes in CeIn₃, which are believed to mediate unconventional superconductivity. Our study paves the way for a quantitative understanding of metallic quantum states such as unconventional superconductivity.

The Kondo lattice model has been key in qualitatively demonstrating how a myriad of correlated quantum matter states emerge[1–8] from the interplay of local and itinerant electrons[8–10]. Beyond the strongly correlated electron materials for which this archetypal model was conceived, it applies to a growing list of novel quantum systems with potential for applications including the electronic transport through quantum dots[11], voltage-tunable magnetic moments in graphene[12], magnetism in twisted-bilayer graphene[13] and in two-dimensional organometallic materials[14], the electronic structure in layered narrow-electronic-band materials[15], electronic resonances of Kagome

¹Laboratory for Neutron and Muon Instrumentation, Paul Scherrer Institute, Villigen, PSI, Switzerland. ²Physik-Institut, Universität Zürich, Winterthurerstrasse 190 CH-8057 Zürich, Switzerland. ³Department of Physics and Astronomy, The University of Tennessee, Knoxville, TN 37996, USA. ⁴School of Physics and Astronomy, University of Minnesota, Minneapolis, MN 55455, USA. ⁵Los Alamos National Laboratory, Los Alamos, NM 87545, USA. ⁶Neutron Scattering Division, Oak Ridge National Laboratory, Oak Ridge, TN 37831, USA. ⁷Laboratory for Neutron Scattering and Imaging, Paul Scherrer Institute, Villigen, PSI, Switzerland. ⁸RIKEN Center for Emergent Matter Science, Wako, Saitama 351-0198, Japan. ⁹Department of Applied Physics, The University of Tokyo, Hongo, Bunkyo-ku, Tokyo 113-8656, Japan. ¹⁰Quantum Condensed Matter Division and Shull-Wollan Center, Oak Ridge National Laboratory, Oak Ridge, TN 37831, USA. ¹¹Present address: Center for Correlated Matter and School of Physics, Zhejiang University, 310058 Hangzhou, China. ✉e-mail: marc.janoschek@psi.ch

metals[16], metallic spin liquid states[17–19] that may even be of chiral character[20], skyrmions in centrosymmetric magnets[21,22] and fully tunable electronic quasiparticles in semiconductor moiré materials[23]. Further, Kondo lattice models have been used to study flat-band materials[24] and predict novel topological states such as topological superconductivity[25] and quantum spin liquid states[26], including the highly sought-after fractional quasiparticles[27]. Despite this continued relevance, quantitative predictions for real materials based on Kondo lattice models remain a formidable computational hurdle.

Metals containing cerium are firmly established model systems for the interplay between itinerant and localized electronic degrees of freedom and are ideal candidates to make progress on this issue. A prototypical case is $CeIn_3$, whose phase diagram as a function of temperature $T$ and hydrostatic pressure $p$ is shown in Fig. 1a[28]. The formation of well-localized magnetic moments occurs due to Ce $4f$ orbitals that are buried close to the nuclei. A weak hybridization with conduction electron bands leads to a long-range magnetic exchange interaction between the moments known as the Ruderman–Kittel–Kasuya–Yosida (RKKY) interaction[29–31] (see Fig. 1b). By increasing the strength of the hybridization, one can screen the magnetic moment through the Kondo effect, leading to a strongly renormalized electronic density of states near the Fermi energy. This heavy Fermi-liquid state borders the magnetically ordered state, from which it is separated by a magnetic quantum phase transition (QPT) that can be accessed via an external control parameter (here: pressure). Interestingly, novel quantum states (here: superconductivity[32]) emerge generically in the vicinity of the QPT suggesting that they are mediated by the associated magnetic quantum critical fluctuations[8]. To understand this emergence near QPTs in a quantitative manner requires a materials-specific microscopic model that incorporates relevant interactions to account for the magnetically ordered state and the resulting fluctuations.

Both for experiment and theory, the challenge in understanding real materials is the extreme energy resolution (~meV) required to capture the inherently small energy scales that emerge in the renormalized electronic state (see Fig. 2a). On the non-magnetic side of the QPT, a pioneering time-of-flight (TOF) neutron spectroscopy study[33], complemented by a subsequent resonant inelastic X-ray (RIXS) scattering measurement[34], on a selected material with a relatively large Kondo interaction ($\approx$60 meV) has only recently achieved quantitative agreement with dynamical mean field theory (DMFT). In contrast, for the magnetic (or superconducting) state of interest here, the relevant energy scale is typically of the order of meV, making this a formidable issue. Materials-specific theoretical investigations of emergent phenomena in $f$-electron materials[35,36] are often limited by the difficulty of validating low-energy effective models derived from complex high-energy input. Consequently, studies of such emergent states of matter are generally restricted to oversimplified models of real materials. Experimentally, in addition to the demand for energy resolution, the pivotal requirement is state-of-the-art momentum-transfer resolution. Notably, the long-range nature of RKKY exchange in real space (Fig. 1b) universally results in extremely sharp magnetic excitations in momentum space, as we elucidate below.

Here we illustrate an approach to reduce a multi-orbital periodic Anderson model (MO-PAM) imbued with materials-specific hopping parameters to a minimal Kondo-Heisenberg lattice model that can form the basis for understanding emergent states of matter in various Kondo lattice materials (Fig. 2a). Using controlled fourth-order perturbation theory and the high combined momentum-transfer and energy resolution of the latest-generation TOF spectrometers, we resolve the full magnetic interaction over the entire range of relevant length scales. We thereby demonstrate that we can determine accurately the emergent energy scales four orders of magnitude smaller than the bare parameters of the initial MO-PAM. In doing so, we critically evaluate the assumptions made to date about heavy-fermion materials.

## Results

Due to a simple cubic structure that facilitates numerical calculations and its characteristic Doniach phase diagram as a function of pressure $p$ (Fig. 1a), $CeIn_3$ is ideally suited to revisit the role of magnetic interactions on a Kondo lattice. As the MO-PAM was originally conceived to explain the localization of shielded electron shells[37] and, in turn, offers a realistic theoretical treatment of coupled charge and spin degrees of freedoms in correlated metals, we begin with a 25-orbital PAM

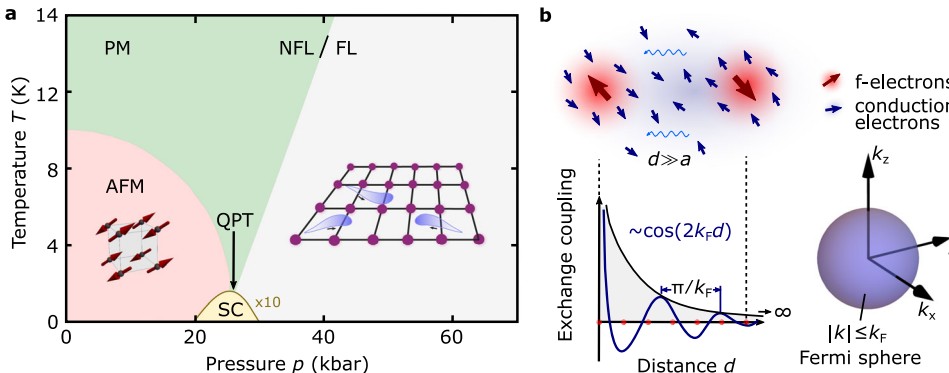

**Fig. 1 | Magnetic interactions and emergent quantum states in a prototypical Kondo lattice. a** The magnetic phase diagram of the heavy electron material $CeIn_3$ as a function of pressure $p$ exemplary of many Kondo lattice materials. The Ruderman–Kittel–Kasuya–Yosida (RKKY) interaction arises when local cerium $4f$ moments polarize the surrounding conduction electron spins, which, in turn, interact with local moments on neighboring Ce sites (see **b**). At ambient pressure this results in antiferromagnetic (AFM) order. However, increasing pressure increases the overlap of neighboring $4f$ electron wave functions, and the hybridization with the conduction electrons. The Kondo effect then leads to the quenching of the local Ce moments, and the formation of strongly-renormalized non-magnetic heavy Fermi liquid (FL). The AFM and FL state are separated by a magnetic quantum phase transition (QPT). The underlying magnetic interactions are therefore not only responsible for the AFM order at ambient pressure, but the associated magnetic quantum fluctuations at the QPT drive the emergence of novel quantum states, in this case, of unconventional superconductivity (SC) and lead to non-Fermi liquid (NFL) behavior. **b** Because the RKKY interaction is mediated via conduction electrons (top panel) it is extremely long-range (bottom panel), compared to the lattice parameter $a$. It is also oscillatory in nature as we illustrate for a spherical Fermi-surface with radius $k_F$ (right panel), which results in a period $\lambda \approx \pi/k_F$ in real space (bottom panel). However, as we demonstrate in Fig. 2, magnetic order on a Kondo lattice is not only mediated by RKKY interactions but additionally requires short-range superexchange and fourth-order particle-particle interactions.

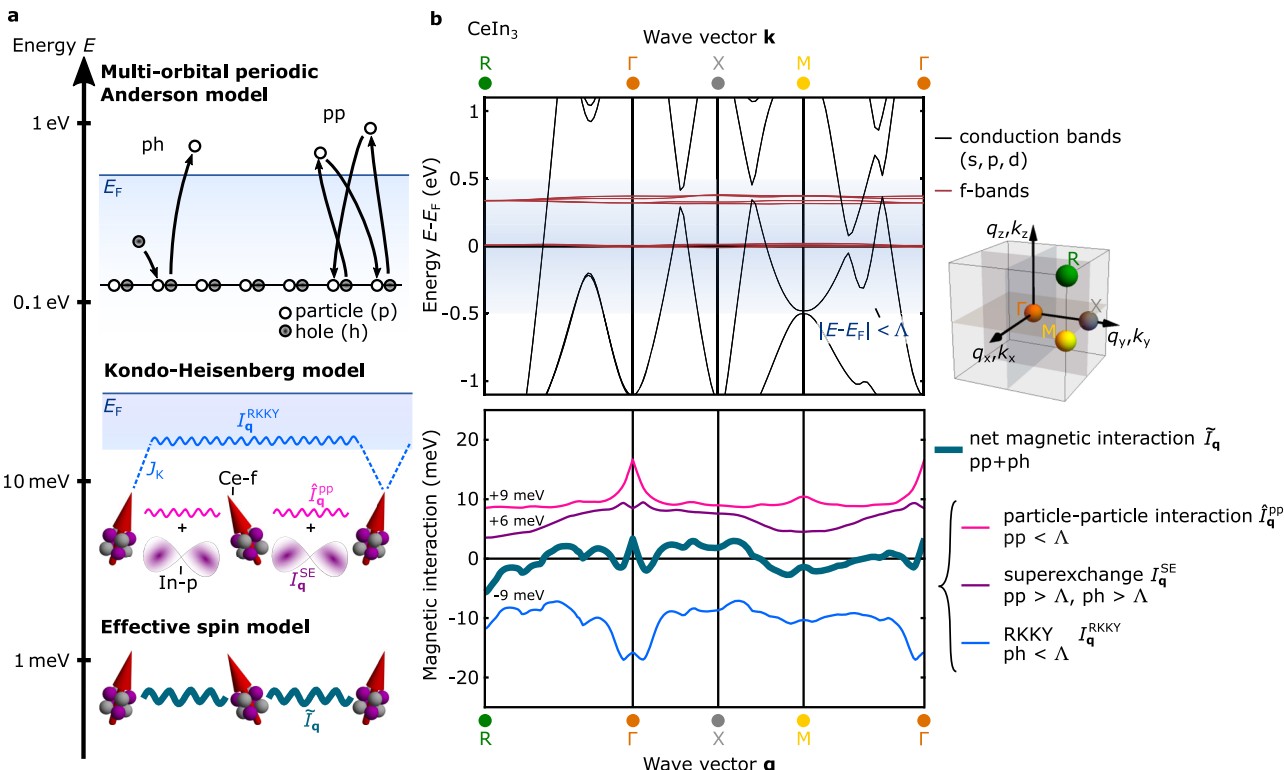

**Fig. 2 | Derivation of the Kondo–Heisenberg model. a** Materials-specific microscopic parameters are contained in the multi-orbital periodic Anderson lattice model (MO-PAM), where $f$-electrons are well localized due to a strong Coulomb repulsion. These states, however, still fluctuate through hybridization with the conduction electron bands. By integrating out the high-energy conduction band states, one arrives at a minimal Kondo + Heisenberg lattice model (KHM). The Kondo interaction contains the spin exchange interaction with the conduction electrons (blue dashed line), while the Heisenberg interaction is a combination of short-range interactions arising from completely filled or completely unoccupied conduction bands (purple $p$-like orbital between the local moments) and a long-range interaction due to particle-particle processes not captured through the Kondo exchange (right process in the MO-PAM sketch, and pink wavy line in the KHM sketch). This Hamiltonian encapsulates the physics of heavy-fermion materials such as CeIn$_3$. Finally, the low-energy conduction electrons can be integrated out to yield an effective spin Hamiltonian between local $f$-moments, which contains both the long-range RKKY interaction from the Kondo exchange (blue wavy line), and the other contributions explained above. **b** Results derived for CeIn$_3$ using this approach are shown. The upper panel shows the electronic bandstructure that serves as input for the MO-PAM along the path RΓXMΓ. The inset on the right illustrates the position of these high symmetry points in the Brillouin zone. The cut-off $\Lambda = 0.5$ eV around the Fermi-energy $E_F$ allows short-range superexchange and long-ranged interactions to be seperated (see text for details). The lower panel shows the various contributions to the net magnetic interaction $\tilde{I}_\mathbf{q}$ in CeIn$_3$ shifted by the indicated energies for visibility. The color of each contribution denotes the corresponding process of the same color in **a**.

for CeIn$_3$,

$$\mathcal{H}_{\text{MO-PAM}} = \sum_{\mathbf{k},s} \epsilon_{\mathbf{k},s} c^\dagger_{\mathbf{k},s} c_{\mathbf{k},s} + \sum_{\substack{i,m,m' \\ \sigma,\sigma'}} h^{\text{SI}}_{m'\sigma';m\sigma} f^\dagger_{i,m'\sigma'} f_{i,m\sigma}$$
$$+ \sum_{\mathbf{k},s,m,\sigma} V_{\mathbf{k},m\sigma} \left( f^\dagger_{\mathbf{k},m\sigma} c_{\mathbf{k},s} + h.c. \right),$$

(1)

where $c^\dagger_{\mathbf{k},s}$ ($c_{\mathbf{k},s}$) is the creation (annihilation) operator of band electrons with wavevector $\mathbf{k}$ and band index $s$, which includes the spin index. $f^\dagger_{i,m\sigma}$ ($f_{i,m\sigma}$) denotes the creation (annihilation) operator of an $f$-electron state with $l_z = m$ ($-3 \le m \le 3$) and spin $\sigma$ on lattice site $i$. The matrix elements $h^{\text{SI}}_{m'\sigma';m\sigma}$ of the single-ion $f$-electron Hamiltonian include the crystal-field coefficients and intra-atomic spin-orbit coupling (see Methods for details). For CeIn$_3$ and related materials, it is well established that the $f^2$ state is energetically considerably less favorable than the $f^0$ excited state[38,39]. Thus, we assume that the on-site repulsive interaction between $f$-electrons is infinitely large, implying that $f^2$ configurations are excluded from the Hilbert space: $f^\dagger_{i,m\sigma} f^\dagger_{i,m'\sigma'} = 0$. The dispersion of the conduction electrons ($\epsilon_{\mathbf{k},s}$) and the hybridization between $f$ and conduction electron states ($V_{\mathbf{k},m\sigma}$) are obtained from a tight-binding fit to the ab initio bandstructure calculation (see Methods and Supplementary Information for details). We find it important to include 18 conduction electron orbitals per

spin (9 In-$p$, 3 In-$s$, 5 Ce-$d$ and 1 Ce-$s$) to account fully for the electronic structure of the conduction bands near the Fermi level $E_F$, and to obtain well localized $f$-orbitals. The spin-orbit coupling and crystal fields of CeIn$_3$ allow that, out of the 14 $f$ states, only the $\Gamma_7$ ground state doublet needs to be included, which is separated by 12 meV from the $\Gamma_8$ quartet excited state[40]. Hence, the energy of the $\Gamma_7$ state ($\epsilon^f_{\Gamma_7}$) is the one remaining free parameter in our model. We further demonstrate in the Supplementary Information that our ab initio band structure calculation is in agreement with the electronic structure determined via angle-resolved photo emission spectroscopy (ARPES) experiments[41].

As the hybridization between the conduction electrons and the $f$-electrons is small, we can use degenerate perturbation theory (see Supplementary Information for details) to derive an effective Kondo-Heisenberg Hamiltonian,

$$\mathcal{H}_{\text{KH}} = \sum_{\mathbf{k},s}' \epsilon_{\mathbf{k},s} c^\dagger_{\mathbf{k},s} c_{\mathbf{k},s} + \sum_{\substack{i,\mathbf{k},\mathbf{k}', \\ \sigma,\sigma',s,s'}}' J^{\mathbf{k}s,\mathbf{k}'s'}_{i,\sigma\sigma'} \tilde{f}^\dagger_{i,\sigma} \tilde{f}_{i,\sigma'} c^\dagger_{\mathbf{k}',s'} c_{\mathbf{k},s} + \sum_{\mathbf{q},\mu,\nu} I^{\mu\nu}_\mathbf{q} S^\mu_\mathbf{q} S^\nu_{-\mathbf{q}'}$$

(2)

where $\tilde{f}^\dagger_{i,\sigma}$ creates an $f$-electron in the lowest energy $\Gamma_7$ doublet state ($\sigma = \{\uparrow, \downarrow\}$) of site $i$, $S^\nu_\mathbf{q}$ is the Fourier transform of the effective spin-1/2 operator $S^\nu_i$ (see Methods). The Kondo coupling is $J^{\mathbf{k}s,\mathbf{k}'s'}_{i,\sigma\sigma'} = \frac{1}{N} e^{i(\mathbf{k}-\mathbf{k}')\cdot\mathbf{r}_i} \tilde{V}_{\mathbf{k},\sigma s} \tilde{V}^*_{\mathbf{k}',\sigma's'} / (E_F - \epsilon^f_{\Gamma_7})$ where $\tilde{V}_{\mathbf{k},\sigma s}$ is the hybridization projected to the $\Gamma_7$ doublet. $\mathcal{H}_{\text{KH}}$ contains effectively only the two

conduction-electron bands that are close to the Fermi energy (the sum $\sum'$ is restricted to band states within a cut-off $\Lambda = 0.5$ eV: $|\epsilon_{\mathbf{k},s} - E_F| \leq \Lambda$ and $|\epsilon_{\mathbf{k}',s'} - E_F| \leq \Lambda$; cf. top of Fig. 2b), as opposed to the 18 conduction bands in the MO-PAM. The remaining bands have been integrated out by including all fourth-order particle-hole and particle-particle processes that give rise to two types of magnetic interactions between the $f$-moments in $I_{\mathbf{q}}^{\mu\nu}$ (last term of Eq. (2)). Notably, in addition to short-ranged superexchange involving particle-hole and particle-particle processes with excited states outside the cut-off $\Lambda$ ($I_{\mathbf{q}}^{SE}$), long-ranged interactions arise from fourth-order particle-particle processes with excited states inside the cut-off ($\hat{I}_{\mathbf{q}}^{(pp)}$) (cf. Fig. 2). This derivation provides the microscopic justification for including a Heisenberg term in model studies of the Kondo lattice (e.g.[42,43]). As we shall see below, it also plays an important role in understanding the magnetically ordered state of CeIn$_3$. Importantly, this minimal Kondo-Heisenberg model still contains the materials-specific information through the dispersion relation of the conduction electrons and the Heisenberg exchange coupling.

To validate this new minimal Kondo-Heisenberg model for CeIn$_3$ against experiments, and to illustrate the importance of the different contributions to magnetic interactions identified above, we further derive the RKKY Hamiltonian from the Kondo lattice term (first two terms of Eq. (2)) via an additional Schrieffer–Wolf transformation (see Supplementary Information)[44,45]. The resulting effective spin Hamiltonian is

$$\mathcal{H}_{\text{spin}} = \mathcal{H}_{\text{RKKY}} + \sum_{\mathbf{q},\mu,\nu} I_{\mathbf{q}}^{\mu\nu} S_{\mathbf{q}}^{\mu} S_{-\mathbf{q}}^{\nu} = \sum_{\mathbf{q},\mu,\nu} \tilde{I}_{\mathbf{q}}^{\mu\nu} S_{\mathbf{q}}^{\mu} S_{-\mathbf{q}}^{\nu}. \tag{3}$$

The effective exchange interaction $\tilde{I}_{\mathbf{q}}$ is then a sum of the RKKY interaction $I_{\mathbf{q}}^{\text{RKKY}}$, the superexchange $I_{\mathbf{q}}^{SE}$ and the particle-particle contribution $\hat{I}_{\mathbf{q}}^{(pp)}$. It turns out to be practically isotropic, $\tilde{I}_{\mathbf{q}}^{\mu\nu} \approx \delta_{\mu\nu}\tilde{I}_{\mathbf{q}}$, as a consequence of the weak influence of the spin-orbit interaction on the $f$–$c$ hybridization amplitudes and the suppression of the $f^2$ magnetic virtual states. In Fig. 2b, we show all resulting contributions along the path R$\Gamma$X M$\Gamma$. The inset on the upper right corner of Fig. 2b shows the position of these high symmetry points in the Brillouin zone that define this path. For comparison with experiment, we note that the antiferromagnetic order of CeIn$_3$ below a Néel temperature $T_N = 10$ K is characterized by a magnetic propagation vector $\mathbf{q}_{\text{AFM}} = (\frac{1}{2}, \frac{1}{2}, \frac{1}{2})$[46] corresponding to the R point (cf. Figs. 1 and 2). Crucially, as shown in Fig. 2b, the RKKY interaction (blue line) typically thought to mediate magnetic order in Kondo lattice materials, has a global minimum near $\Gamma = (0, 0, 0)$ and only a local minimum at R, demonstrating that it is not responsible for the onset of the observed order. Instead, the superexchange $I_{\mathbf{q}}^{SE}$ (purple line) exhibits a global minimum at the R point.

Finally, the particle-particle contribution near $E_F$ ($\hat{I}_{\mathbf{q}}^{(pp)}$, pink line) is mostly flat, but is characterized by a pronounced, cusplike maximum at $\Gamma$. It is then essential to note that the global minimum of the net interaction $\tilde{I}_{\mathbf{q}}$ (bold cyan line) at the ordering wavevector R arises from the combination of two effects: a compensation between low-energy particle-particle and particle-hole contributions around the $\Gamma$ point and the short-range antiferromagnetic superexchange interaction generated by the high-energy processes in both channels.

To go beyond the magnetic ground state and to test the theoretically calculated exchange interaction $\tilde{I}_{\mathbf{q}}$ quantitatively against experiment, we have carried out high-resolution neutron spectroscopy to measure the dispersion of the magnons in CeIn$_3$, which is determined by the magnetic interactions derived above. Notably, the dispersion is given by $E_{\mathbf{q}} = S\sqrt{\left(\tilde{I}_{\mathbf{q}_{\text{AFM}}} - \tilde{I}_{\mathbf{q}}\right)\left(\tilde{I}_{\mathbf{q}_{\text{AFM}}} - \tilde{I}_{\mathbf{q}_{\text{AFM}}+\mathbf{q}}\right)}$ with $S = 1/2$

and is shown as the solid blue line in Fig. 3a, which presents an overview of our results at $T = 1.8$ K. The key signature predicted by our model is a extremely dispersive magnon around the R point arising due to the steep local minimum of the long-ranged RKKY contribution; this was not identified in a previous study with modest resolution by Knafo et al., who instead reported a large magnon gap of more than 1 meV[40]. In contrast, the magnon dispersion determined from our neutron spectroscopy data as well as the observed magnetic intensity are in quantitative agreement with the calculated dispersion and intensity, respectively (see Fig. 3). Here the neutron intensities in Fig. 3 are expressed as the dynamic magnetic susceptibility $\tilde{\chi}''(\mathbf{Q}, E)$ and were converted to absolute units of $\mu_B^2 \text{meV}^{-1}$ for comparison to theory. In particular, the energy and momentum transfer cuts through the magnon spectrum shown in Figs. 3c and d, respectively, demonstrate the excellent agreement between experiment and theory. Figure 4 showcases the steep magnon dispersion in a small region around the R point, which could only be uncovered by the most modern spectrometers (see Methods). Here panels **a1, b1,** and **c1** show slices through the R point along the three cubic high-symmetry directions and confirm that the magnon gap is either absent or substantially smaller than the experimental resolution ($\Delta E = 106$ $\mu$eV) as we show in detail in the Supplementary Information. A single fit parameter, $\epsilon_{\Gamma_7}^f = 12.009$ eV, reproduces the experimental dispersion both with regard to the bandwidth and the magnon velocity. This showcases that our microscopic model, which starts with an ab initio bandstructure calculation with energy scales of 10 eV, is able to predict magnetic interactions on the order of meV. In addition to being quantitatively accurate, our calculations are robust against small changes of the chemical potential of the order of 10 meV (resolution of our band structure calculation), which modify the slope and bandwidth of the magnon dispersion by less than 1

Finally, we consider the role of short-range Heisenberg superexchange. Although the long-range RKKY interaction is widely credited with mediating magnetic order in Kondo lattice materials[8], short-range superexchange interactions are commonly employed to fit the observed magnon dispersion[47–52] and have been also used for CeIn$_3$[40]. As expected short-range superexchange allows us to model the magnon dispersion at high energy and the zone boundary well (cf. Fig. 3). Besides providing a microscopic justification for including short-range superexchange interactions, our model also demonstrates that they are not sufficient to explain the magnons near the magnetic zone center. We quantify this statement via the dimensionless parameter $\eta$ that describes ratio of the magnon velocity $v$ to the magnon bandwidth $W$. The experimental magnon bandwidth is $W_{\text{exp}} = 2.75(3)$ meV, as determined from the energy cuts in Fig. 3b. The experimental magnon velocity, $v_{\text{exp}} = 38.6(8)$ meV/r.l.u., was inferred from linear fits to the measured dispersion near to the R point along the different high-symmetry directions and averaged over all directions (black dashed lines in Fig. 4 and Supplementary Information). The parameter $\eta_{\text{exp}} = 2.23(6)$ derived from our experiments compares favorably with $\eta_{\text{MO–PAM}} = 2.27(5)$ computed from our model. In contrast, fits of the magnon dispersion with a single exchange constant $J_1$ result in $\eta_{J_1} = 0.58$, which stems from a substantial underestimation of the magnon velocity near the R point (cf. light green line in Fig. 4). We note that even adding a large number of additional higher-order exchange terms does not allow the large observed magnon velocity to be modeled accurately (see Supplementary Information), as was noted in previous studies[47,51].

## Discussion

Our study demonstrates that it is now possible to derive a minimal microscopic Kondo lattice model for a real material starting from ab initio electronic bandstructure calculations starting at energy scales of eV. This enables us to reproduce quantitatively the magnetic order and

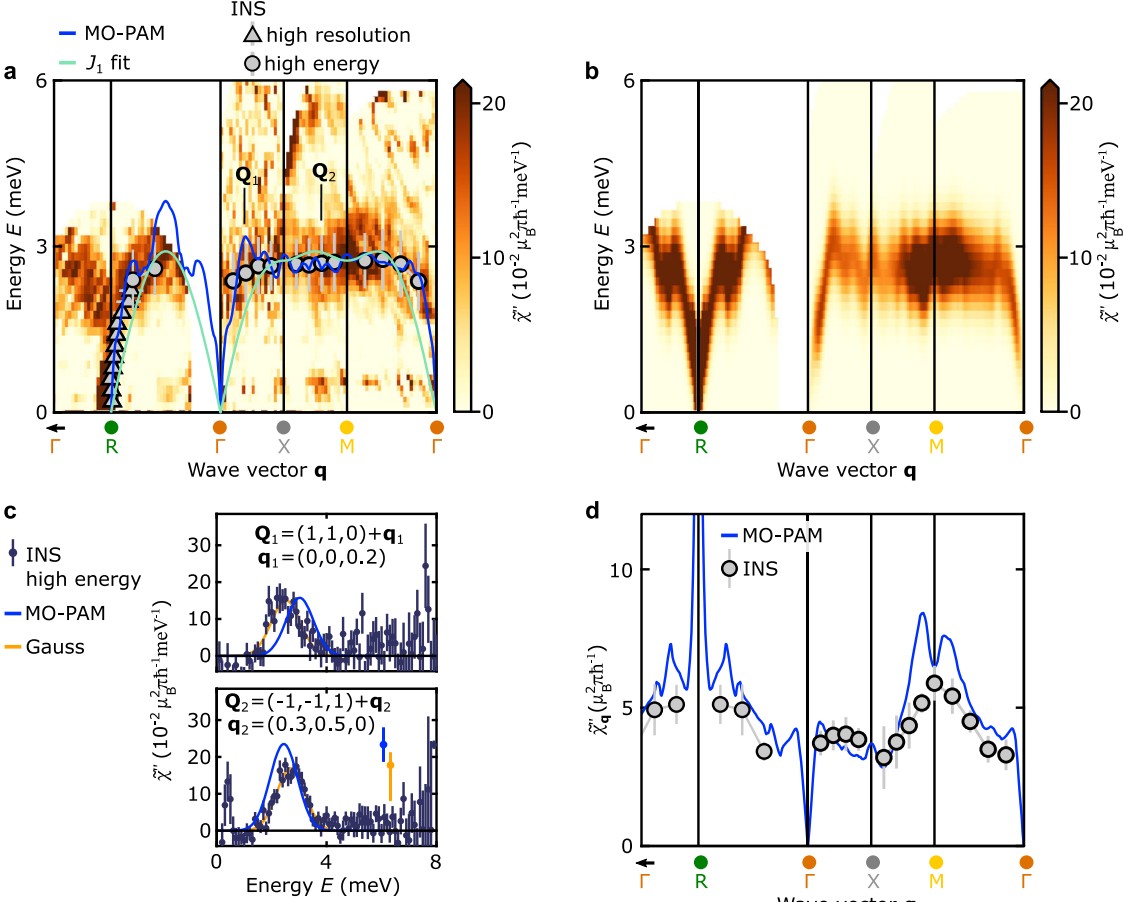

**Fig. 3 | Calculated and measured magnon dispersion and dynamic magnetic susceptibility in the antiferromagnetic state of CeIn₃.** Panels **a** and **b** show the imaginary part of the dynamic magnetic susceptibility $\tilde{\chi}''(\mathbf{Q}, E)$ obtained by our experiments and calculated via the MO-PAM model, respectively. Solid lines denote the magnon dispersion on the path $\gamma = R\Gamma X M\Gamma$ as calculated based on the effective spin interaction $\tilde{I}_{\mathbf{q}}$ [Eq. (3)] derived from the MO-PAM (blue solid line, see text for details), a Heisenberg model with a single nearest-neighbor exchange constant $J_1$ (light green line), and the dispersion inferred from our neutron spectroscopy measurements (symbols). We refer to the inset on the upper right corner of Fig. 2b for the position of the high symmetry positions $R$, $\Gamma$, $X$, $M$, and $\Gamma$ in the Brillouin zone. Experimental data points and error bars represent the location and standard deviation of the Gaussian peaks inferred from neutron spectroscopy data sets with incident neutron energies 3.315 meV (high-resolution, triangles) and 12 meV (high-energy, circles), by means of constant-energy and constant-momentum transfer

cuts, respectively. All cuts investigated and the corresponding fits are shown in the Supplementary Information. The color-scale provides $\tilde{\chi}''(\mathbf{Q}, E)$ in absolute units of $\mu_B^2 \mathrm{meV}^{-1}$ along the path $\gamma$, as inferred from neutron spectroscopy data taken in the high-energy setting. For details see Methods. **c** Typical constant-momentum-transfer cuts through $\tilde{\chi}''(\mathbf{Q}, E)$ as illustrated for the two positions $\mathbf{Q}_1$ and $\mathbf{Q}_2$ on the paths $X\Gamma$ and $XM$, respectively. Square symbols denote high-intensity neutron spectroscopy data and the blue solid lines MO-PAM calculations. Error bars represent the statistical error. The orange line corresponds to a Gaussian fit to the INS data. The orange and blue bars denote the systematic error in the conversion of the neutron intensity to absolute units and uncertainties in the calculation of the MO-PAM intensities, respectively (see Supplementary Information for details). **d** The integral of $\tilde{\chi}''(\mathbf{Q}, E)$ over energy, denoted $\tilde{\chi}''_{\mathbf{Q}}$. Circle symbols and the blue solid line correspond to $\tilde{\chi}''_{\mathbf{Q}}$ from high-intensity neutron data and MO-PAM calculations, respectively. Error bars represent the statistical error.

all relevant magnetic interactions in the prototypical Kondo lattice CeIn₃ at energy scales 10,000 times smaller. Thus, it provides a tractable path for the accurate calculation of the low-energy spin excitations that arise due to strong electronic correlations, and are believed to mediate the emergence of novel quantum phases (cf. Fig. 1). Due to the long-range nature of the RKKY interaction, these magnetic soft modes are remarkably steep. This is also borne out by neutron scattering studies on further $f$-electron materials, which highlight the broad relevance of our results. Either steep magnon dispersions were observed directly[47,51,53] or this key feature is concealed due to inadequate resolution, resulting in reports of potentially spurious magnon gaps[48,52,54] similar to CeIn₃[40]. We note that the ability to resolve these steep low-energy spin excitations also ushers in high-resolution TOF spectroscopy as a complementary technique to access the electronic band structure of magnetically ordered heavy-fermion materials.

A further remarkable insight revealed by our calculations is that, in addition to the RKKY interaction, contributions from the particle-

particle channel are equally crucial to quantitative description of the magnetic order and the magnon spectrum. In turn, our approach resolves several puzzles concerning magnetic order in heavy-fermion materials. First, it is consistent with neutron scattering studies of the magnon spectrum of various heavy fermion-materials, where the dispersion towards the zone boundary is well explained using short-range interactions[47–49,49–52]. Second, the presence of short-range interactions highlights why a large collection of heavy fermion materials exhibit commensurate AFM order[46,48,55–57], even though RKKY interactions arising from generic Fermi surfaces typically favor incommensurate order. Finally, the apparent lack of 4$f$-based metallic ferromagnets is now understood quite simply by the presence of particle-particle interactions in metals, which generically disfavor the ferromagnetic ground state.

In summary, the combination of our MO-PAM approach with TOF spectroscopy establishes a straightforward recipe to obtain a quantitative, yet relatively simple, effective Kondo-Heisenberg Hamiltonian,

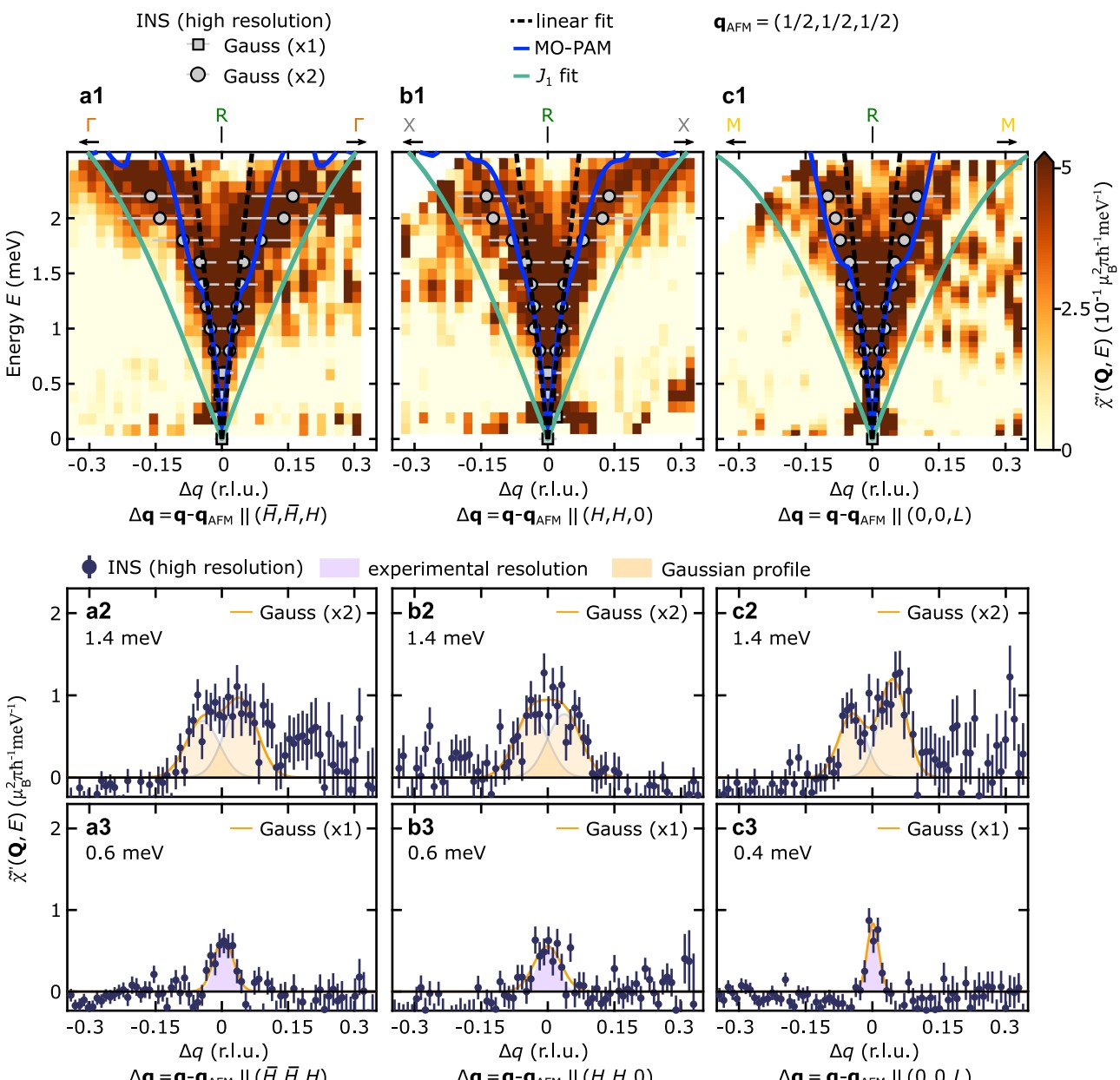

**Fig. 4 | Signature of long-range RKKY interactions in CeIn₃. a1–c1** Magnon dispersion near the magnetic zone center R along the respective lines RΓ, RX, and RM. The color-coding reflects the imaginary part of the dynamic susceptibility, $\bar{\chi}''(\mathbf{Q},E)$, recorded with incident neutron energy $E_i = 3.315$ meV (see Methods for details). Squares and circles correspond to the locations of Gaussian profiles inferred from constant-energy cuts with one and two Gaussian profiles, respectively. All cuts investigated and the corresponding fits are shown in the Supplementary Information. The solid blue and light green lines denote the magnon dispersions calculated from our microscopic model and a fit of a short-range Heisenberg model with a single nearest-neighbor exchange constant $J_1$ to our data, respectively. The dashed black line is a linear fit of the magnon dispersion close to the zone center used to extract the magnon velocity. **a2–c2** At higher energies, where the splitting is larger than the FWHM of the experimental resolution, the profiles of constant-energy cuts were fitted by two Gaussian profiles. **a3–c3** At energies below 0.6 meV for the RΓ and RX direction and below 0.4 meV for the RM direction, the splitting is restricted by the experimental resolution as indicated by the shaded profile. Error bars in panels **a2–c2** and **a3–c3** represent the statistical error.

which for CeIn₃ includes only 2+1 orbitals (two conduction electron bands and one $f$ level). In turn, studying this effective model as a function of pressure has the promising prospect of establishing the emergence of unconventional superconductivity in CeIn₃ quantitatively. Notably, the short-range interaction promotes a local-moment magnetic state relative to the bare Kondo temperature, which may fundamentally alter the nature of the magnetic QPT observed under pressure[58]. Adding the $\Gamma_8$ $f$-state to our calculation should equally allow us to reproduce the magnetic anisotropy emerging as a function of magnetic field[59]. Considering the ever-increasing computational power available, our approach is equally in reach for more complex materials with lower symmetry and more orbitals, which will allow us to unlock the microscopic understanding of a large number of quantum matter states with functional properties[8]. Further, our discovery that both short- and long-range interactions are key to understanding magnetic order in heavy fermion materials offers a straightforward explanation of their rich magnetic phase diagrams, where changing the balance of interactions by applying external tuning parameters allows us to select distinct magnetic ground states. Similarly, we anticipate that the combination of short- and long-ranged interactions will turn out to be a generic feature of metallic systems whose starting point is the periodic Anderson model. Finally, our study paves the way for ab

initio modeling of quantum systems described by Kondo lattices and therefore suggests an avenue beyond a phenomenological description of the ground states of their strongly correlated electron systems.

## Methods

### Multi-orbital periodic Anderson model

The input parameters for the multi-orbital periodic Anderson model (MO-PAM) presented in Eq. (1) specific to CeIn$_3$ were obtained by deriving a tight-binding model based on density functional theory (DFT) that accurately captures the electronic structure of the conduction bands near the Fermi level $E_F$, and yields the hybridization between these bands and the 4$f$-orbitals. The underlying DFT band structure calculations were performed using the QUANTUM ESPRESSO package[60] and fully relativistic projector augmented-wave (PAW) pseudopotentials with the Perdew-Burke-Ernzerhof (PBE) exchange-correlation functional, which are available in PSlibrary[61]. A realistic tight-binding Hamiltonian with 50 Wannier functions (25 orbitals times 2 for spin) was constructed using the Wannier90 package[62]. Details for all steps are provided in the Supplementary Material.

The low-temperature magnetic properties of CeIn$_3$ at zero field are dominated by the low-energy Ce $4f\Gamma_7$ doublet that results from diagonalizing the single-site $f$-electron Hamiltonian [second term of Eq. (1)],

$$h^{SI}_{m'\sigma';m\sigma} = \epsilon^f \delta_{\sigma,\sigma'}\delta_{m,m'} + B_{m,m'}\delta_{\sigma,\sigma'} + \lambda\zeta_{m'\sigma';m\sigma},\quad(4)$$

which includes the crystal field coefficients $B_{m,m'}$ and the *intra-atomic spin-orbit coupling* $\lambda$ ($\zeta_{m'\sigma';m\sigma} = \delta_{m',m}\delta_{\sigma',\sigma}m\sigma/2 + \delta_{m',m+\sigma}\delta_{\sigma',-\sigma}\sqrt{12-m(m+\sigma)}/2$). The resulting $\Gamma_7$ doublet reads

$$|\Gamma_7;+\rangle \equiv |\uparrow\rangle = \sqrt{\frac{1}{6}}\left|j=\frac{5}{2},m_j=\frac{5}{2}\right\rangle - \sqrt{\frac{5}{6}}\left|j=\frac{5}{2},m_j=-\frac{3}{2}\right\rangle,\quad(5a)$$

$$|\Gamma_7;-\rangle \equiv |\downarrow\rangle = \sqrt{\frac{1}{6}}\left|j=\frac{5}{2},m_j=-\frac{5}{2}\right\rangle - \sqrt{\frac{5}{6}}\left|j=\frac{5}{2},m_j=\frac{3}{2}\right\rangle.\quad(5b)$$

By taking the limit of infinite intra-atomic $f$-$f$ Coulomb repulsion which eliminates the $f^2$ configurations, we project $\mathcal{H}_{MO-PAM}$ into the low-energy subspace generated by the $\Gamma_7$ doublet and obtain the periodic Anderson model

$$\mathcal{H}_{PAM} = \sum_{\mathbf{k},s}\epsilon_{\mathbf{k},s}c^\dagger_{\mathbf{k},s}c_{\mathbf{k},s} + \epsilon^f_{\Gamma_7}\sum_{i,\sigma}\tilde{f}^\dagger_{i,\sigma}\tilde{f}_{i,\sigma} + \sum_{\mathbf{k},\sigma,s}\left(\tilde{V}_{\mathbf{k},\sigma s}\tilde{f}^\dagger_{\mathbf{k},\sigma}c_{\mathbf{k},s} + h.c.\right),\quad(6)$$

where the constrained operators $\tilde{f}^\dagger_{i,\sigma}$ ($\tilde{f}_{i,\sigma}$) create (annihilate) an $f$-electron in the $\Gamma_7$ doublet with $\sigma = \{\uparrow,\downarrow\}$ and $\tilde{f}^\dagger_{i,\sigma}\tilde{f}^\dagger_{i,\sigma'} = 0$, $\epsilon^f_{\Gamma_7}$ is the energy of the $\Gamma_7$ states, and $\tilde{V}_{\mathbf{k},\sigma s}$ is the hybridization between the $\Gamma_7$ doublet and the conduction electron states ($1 \leq s \leq 36$).

By treating the small hybridization $\tilde{V}_{\mathbf{k},\sigma s}$ as a perturbation, the periodic Anderson model can be reduced to the effective Kondo–Heisenberg Hamiltonian shown in Eq. (2). Furthermore, for strongly localized $f$-electrons, the Kondo lattice model can be further reduced to the RKKY Hamiltonian via second order degenerate perturbation theory in the Kondo interaction (see Supplementary Information for details). The final effective spin Hamiltonian $\mathcal{H}_{spin}$ is presented in Eq. (3), where the pseudo spin-$\frac{1}{2}$ operator is defined as $\mathbf{S}_i \equiv \frac{1}{2}\tilde{f}^\dagger_{i,\alpha}\boldsymbol{\sigma}_{\alpha\beta}\tilde{f}_{i,\beta}$ ($\sigma^\nu$ are the Pauli matrices with $\nu = x,y,z$), and its Fourier transform is defined as $S^\nu_\mathbf{q} = \frac{1}{N}\sum_i e^{i\mathbf{q}\cdot\mathbf{r}_i}S^\nu_i$. $N$ is the total number of Ce atoms on the lattice (we take $N \to \infty$ in the following calculation).

### Sample preparation

To overcome the high absorption of cold neutrons by indium, plate-like single crystalline samples of CeIn$_3$ were grown by the indium self-flux method and polished to a thickness of around 0.7 mm. To maximize the total scattering intensity 24 pieces were carefully coaligned on an aluminum sample-holder using a hydrogen-free adhesive (CYTOP) (see Sec. II.A of the Supplementary Information).

### Neutron spectroscopy

Inelastic neutron scattering was carried out at the cold neutron chopper spectrometer CNCS at ORNL[63]. For the measurements, the crystal array was oriented in such a way that the crystallographic $[1\bar{1}0]$ axis was vertical. Momentum transfers of neutrons are given in the reference frame of the sample by means of $\mathbf{Q} = H\mathbf{b}_1 + K\mathbf{b}_2 + L\mathbf{b}_3$, where $H, K$, and $L$ denote the Miller indices and $\mathbf{b}_\nu = \frac{2\pi}{a}\hat{\mathbf{a}}_\nu$ ($\nu = 1, 2, 3$) represent primitive translation vectors of the reciprocal cubic lattice ($a = 4.689$ Å). Throughout the manuscript and the SI, when stating components of momentum transfers the reciprocal lattice unit ($1 \cdot$ r.l.u. $:= \frac{2\pi}{a}$) is omitted. To a given momentum transfer $\mathbf{Q}$ (upper-case letter), the reduced momentum transfer that equals the equivalent reciprocal space position in the cell $0 \leq q_\nu < 1$, is given by $\mathbf{q} = q_1\mathbf{b}_1 + q_2\mathbf{b}_2 + q_3\mathbf{b}_3$ (labeled by a lower-case letter). $\Delta q$ denotes the modulus of $\Delta\mathbf{q}$. Reciprocal-space distances are also given in units $1 \cdot$ r.l.u. $:= \frac{2\pi}{a}$.

Time-of-flight neutron spectroscopy was performed with two different incident neutron energies. High-resolution experiments with incident neutron energy $E_{i,1} = 3.315$ meV ($\Delta E = 106\ \mu$eV) permitted the study of the steep magnon dispersion in the vicinity of the reciprocal space position $\mathbf{Q}_0 = \left(-\frac{1}{2}, -\frac{1}{2}, \frac{1}{2}\right)$, which represents the R point. A "high-energy" setting with incident neutron energy $E_{i,2} = 12$ meV was performed to determine the magnon dispersion across the entire Brillouin zone. Data were collected in terms of a so-called Horace scan[64], where the crystal is rotated around the vertical axis. In combination with the CNCS detector, which has large horizontal (from -50 to 140 degrees) and vertical ($\pm 16$ degrees) coverage, this allows one to obtain four dimensional data sets covering all three momentum transfer directions and energy transfer. Detailed information is provided in Sec. II of the Supplementary Information.

For Figs. 3, 4 the neutron intensity recorded in our experiments has been expressed as the imaginary part of the dynamic magnetic susceptibility $\tilde{\chi}''(\mathbf{Q}, E)$ in absolute units of $\mu_B^2$meV$^{-1}$ to enable direct comparison with the calculations of the dynamic magnetic susceptibility via the Multi-Orbital Periodic Anderson Model (see above). For this purpose, the recorded neutron intensities have been corrected for neutron absorption, put on an absolute scale via comparison to the well-known incoherent scattering of the sample, and finally background subtracted via data sets obtained below $T_N$ (for details of this procedure, we refer to the Supplementary Information). Further, the resulting corrected and normalized magnetic intensity was then corrected by the Bose factor accounting for the thermal population of magnon states, and subsequently divided by the square of the magnetic form-factor for Ce$^{3+}$ as well as the ratio $|\mathbf{K}_f|/|\mathbf{K}_i|$, whereby $\mathbf{K}_i$ and $\mathbf{K}_f$ denote the wavevectors of incident and scattered neutrons, respectively. For further information, we refer to the Supplementary Information. In the following, we additionally detail the integration ranges for the data shown in Figs. 3, 4. The data in Fig. 3a on the paths $\Gamma$R, R$\Gamma$, $\Gamma$X, XM, and M$\Gamma$ were taken from the reciprocal space lines between the $\mathbf{Q}$-positions $(-1, -1, 1)$ and $\left(-\frac{1}{2}, -\frac{1}{2}, \frac{1}{2}\right)$, between $\left(-\frac{1}{2}, -\frac{1}{2}, \frac{1}{2}\right)$ and $(0, 0, 0)$, between $(1,1,0)$ and $\left(1, 1, \frac{1}{2}\right)$, between $(-1, -\frac{1}{2}, 1)$ and $\left(-\frac{1}{2}, -\frac{1}{2}, 1\right)$, and between $\left(\frac{1}{2}, \frac{1}{2}, 0\right)$ and $(1, 1, 0)$, respectively. The intensity was integrated within a distance of $\pm 0.17$ r.l.u. along the two $\mathbf{Q}$-directions that are perpendicular to the path $\Gamma$XM and a distance of $\pm 0.09$ r.l.u. along the two $\mathbf{Q}$-directions that are perpendicular to the paths R$\Gamma$ and M$\Gamma$. For the energy cuts shown Fig. 3b the intensity was integrated within a distance of $\pm 0.17$ r.l.u. along three

different reciprocal space directions. For the reduced momentum transfer cuts and slices shown in Fig. 4, the intensity was integrated within a distance of 0.08 r.l.u. along the two reciprocal space directions that are perpendicular to the reduced momentum transfer **q**. For the cuts in the lower panels, intensity was integrated over energies ± 0.1 meV.

We note that to resolve the sharp magnon dispersion high momentum resolution is required in addition to extreme energy resolution. Although triple-axis neutron spectroscopy is often credited with the best combined momentum-energy-transfer resolution, it fails to properly identify the dispersion even when used with the best resolution due to so-called Currat-Axe spurions[65]. To demonstrate this we have carried out additional measurements on the multiplexing triple axis spectrometer CAMEA at Paul Scherrer Institute[66] using the identical sample and incident energies $E_i$ = 4.15 meV, 4.8 meV, and 5.5 meV. Details of these measurements and the spurious features close to the zone center are shown in the Supplementary Information. This highlights the importance of modern TOF spectroscopy for the investigation of strongly correlated metals.

## Data availability
The datasets generated during and/or analyzed during the current study have been deposited on the ONCat platform of Oak Ridge National Laboratory (ORNL) under the following digital object identifier: 10.14461/oncat.data.64d61647fd6850c0afce4da3/1994514 (https://doi.org/10.14461/oncat.data.64d61647fd6850c0afce4da3/1994514). Currently, the DOI-minting functionality in ONCat is still in beta and behind a firewall and you need to be on ORNL's network to access it. This can be achieved going to https://analysis.sns.gov/ and registering an account, signing and starting a session. As soon as the DOI-minting functionality is publically available the data will directly available via the link above. The processed neutron spectroscopy data shown in the manuscript are available at the Zenodo database under accession code digital object identifier: 10.5281/zenodo.10146787 (https://zenodo.org/records/10146787)[67].

## Code availability
All code for the absorption correction of the neutron spectroscopy data is available in the Zenodo database under accession code digital object identifier: 10.5281/zenodo.10147126 (https://zenodo.org/records/10147126)[68].

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

## Acknowledgements

M.J. would like to thank Georg Ehlers for useful discussions concerning the setup of the CNCS experiment and acknowledges fruitful discussions with Bruce Normand. Work at Los Alamos National Laboratory was performed under the U.S. DOE, Office of Science, BES project "Quantum Fluctuations in Narrow Band Systems". This research used resources at the Spallation Neutron Source, a DOE Office of Science User Facility operated by the Oak Ridge National Laboratory. WS is supported through funding from the European Union's Horizon 2020 research and innovation programme under the Marie Sklodowska-Curie grant agreement No 884104 (PSI-FELLOW-III-3i). Z.W. was supported by funding from the Lincoln Chair of Excellence in Physics. During the writing of this paper, Z.W. was supported by the U.S. Department of Energy through the University of Minnesota Center for Quantum Materials, under Award No. DE-SC-0016371. FR thanks the hospitality of the University of Tokyo. Y.N. and R.A. acknowledge funding through Grant-in-Aids for Scientific Research (JSPS KAKENHI, Japan) [Grant No. 20K14423 and 21H01041, and 19H05825, respectively] and "Program for Promoting Researches on the Supercomputer Fugaku" (Project ID:hp210163) from MEXT, Japan.

## Author contributions

C.D.B., F.R., and M.J. conceived the study. Y.N., R.A., and F.R. carried out the electronic band structure calculations. Z.W., E.A.G., and C.D.B. derived the multi-orbital periodic Anderson model and Kondo–Heisenberg model. D.M.F. and M.J. designed the TOF experiments. W.S., D.G.M., and M.J. designed the CAMEA experiments. D.M.F., W.S., A.P., J.L., S.F., J.V., D.G.M., C.N., and M.J. carried out the experiments. J.V. comissioned and set up the low-temperature sample environment for CAMEA. N.H.S. and E.D.B. synthesized and characterized the samples. D.M.F. assembled the single-crystal mosaic. W.S. and M.J. analyzed the data. W.S., Z.W., C.D.B., F.R., and M.J. interpreted the results. W.S., Z.W., C.D.B., F.R., and M.J. wrote the paper with input from all the authors.

## Competing interests

The authors declare no competing interests.
