## [Peer Review File · Nature Communications]

Reviewers' Comments:

Reviewer #1:

Remarks to the Author:

Referee's report on A microscopic Kondo lattice model for the heavy-fermion antiferromagnet CeIn₃ by Simeth et al.

This paper reports a combined theoretical and experimental study on CeIn₃, the parent compound of Ce-based heavy fermion superconductors including CeCoIn₅. CeIn₃ itself is also a heavy fermion superconductor under pressure. Overall, the paper is well-written, and a substantial amount of work has been put into the paper by the authors. The key conclusion of the work is that a microscopic Kondo lattice model starting from ab initio electronic band structure calculations can quantitatively account for the magnetic order and excitations. Although these results are somewhat interesting, I don't see that there will be tremendous excitement about these results as expected for a Nature Communication publication. I reached this conclusion for the following reasons.

1. The advancements in the experimental part of the work are rather limited. I was not aware of Ref. [39]. But after carefully studying the neutron scattering work of this paper, and comparing that with Ref. [39], there is really not that much new in the present paper. Sure, the modern measurements have higher resolution, and thus are able to resolve spin excitations near the R point. But the overall spin excitations are consistent with the ref. [39].
2. The authors argue that modern calculations can quantitatively predict magnetic exchange interactions and spin excitations. If this is conclusively demonstrated, then the paper would be quite interesting. Given that the experimental results are not in absolute units, only the dispersion part of the spectral is used to compare with the calculation.
3. Since modern neutron scattering experiments should be able to provide absolute magnetic scattering intensity by using a vanadium standard normalization, checking magnetic scattering with acoustic phonon modes, and properly correcting for the large In neutron scattering absorption, a careful neutron scattering experiment on the system should in principle be able to determine the magnetic scattering cross section in absolute units.
4. As the authors argue that their calculation can quantitatively account for the spin excitations of CeIn₃ using a microscopic Kondo lattice model from ab initio electronic bandstructure calculations, I would assume that they can also do this in absolute units. If these calculations are in absolute units, to compare with neutron scattering data in absolute units, I would then trust the results much more, and be willing to change my mind about the recommendation.
5. The work is quite interesting, but the field of heavy fermion magnetism is rather small without too many workers. I would assume that citations for this work will also be limited compared with other active fields.
6. The authors find in the concluding paragraph that both short- and long-range interactions are important for understanding magnetism in heavy Fermion materials. But this is similar to saying both itinerant electrons and localized moments are important for magnetism in many other correlated electron materials, including iron-based superconductors. Such a general conclusion does not help in solving specific problems of understanding spin excitations in heavy fermion superconductors. For example, CeCoIn₅ superconductor has a strong spin collective excitation coupled with superconductivity (resonance), but there is no understanding on the microscopic origin of the resonance. Since CeIn₅ is also a superconductor under pressure, if the authors' theory can make any qualitative or quantitative statement about the origin of superconductivity, which band is important and why magnetism might be central to the origin, the paper would find a much bigger audience and be appreciated by a much broader community. As written, the paper is only interesting for people working on CeIn₅ family of heavy fermions. Since the community is very limited, the paper will have a limited impact on the community.

Overall, I feel that this is a nice piece of work but does not reach to the level of Nature Communications publication. If the authors can redo the data analysis and calculation for absolute intensity comparison, it might warrant a high-profile publication.

Reviewer #2:
Remarks to the Author:
Authors and Editors,

My report is structured to discuss the six topics outlined in the Nature Communications review system, as well as a couple of specific comments about the manuscript at the end. I am happy to discuss any of these comments as necessary if something is not clear. My report is below:

1) Are the results noteworthy?

Quantitative theoretical predictions in correlated electron materials are notoriously difficult to make. Using the well studied cubic Kondo lattice material CeIn₃ as a test, the authors start from a periodic Anderson model and reduce the relevant Hamiltonian to a more simple Kondo-Heisenberg model. Their model reproduces the known magnetic excitations reasonably well and predicts the existence of an unknown magnetic soft mode. They subsequently measured this mode with new, high resolution inelastic neutron scattering experiments. While not perfect, they find decent quantitative agreement with the predicted dispersion of this new mode. This is noteworthy, given that the Kondo lattice model, while applicable to a wide variety of systems, is typically thought of as a successful description if it yields qualitative predictions.

2) Will the work be of significance to the field and related fields? How does it compare to the established literature?

Given the difficulty of making quantitative predictions in correlated electron materials, this work will be of significance to the community. The authors' main accomplishment is that they predicted a significant deviation from the accepted knowledge of the magnetic excitations in CeIn₃, a very well studied material. They then observed this deviation and found decent enough agreement with their theory when they looked with sufficient energy and momentum resolution on a modern time-of-flight neutron spectrometer. The effective low energy theory that made this prediction was developed starting from a high energy Hamiltonian and electronic structure considerations. Making reliable quantitative predictions in this fashion is highly unusual and will surely be of interest to a broad audience.

I note that their ultimate theory of the low energy excitations relies on the fact that CeIn₃ has relatively simple symmetries and is extremely well studied at this point. I question how effective this approach will be on systems that aren't as well studied or ones with more complexity. The authors suggest in their concluding paragraph that the ever increasing computational power available may lead to similar theories for other materials. While it's certainly true that more computing power will allow us to do more computationally, the low energy theories that could be derived with these methods are only as good as the starting inputs and the ability to simplify the Hamiltonians, both of which seem to be greatly assisted here by the very nature of CeIn₃.

Having said that, I don't want to take away from their main accomplishment. This will certainly be of interest, even if I am a bit more skeptical than the authors about how broadly applicable it might be.

I make a specific comment on the established literature in section 4 below.

3) Does work support conclusions and claims, or is additional evidence needed?

Assuming the derivation of their low energy theory is robust, and that the authors can address the other concerns outlined here, then I think the work does generally support their conclusions. My reservations about the broad applicability of the technique presented (at least in the near or medium term) are outlined above. To overcome this limitation would require a lot more theory work and experiments on different materials. That would clearly be beyond the scope of this manuscript, so I don't think my general reservations should preclude publication.

However, in this spirit, I ask the authors to reconsider one of their sentences towards the end of the paper: "We note that the ability to resolve these steep low-energy spin excitations also ushers

in high-resolution TOF spectroscopy as a novel technique to verify the electronic band structure of magnetically ordered heavy-fermion materials." As I said, I'm not as convinced as they are about how this can be applied to other systems in the immediate future.

I will leave it to other reviewers to comment on the more technical theory aspects.

4) Are there any flaws in data analysis, interpretation and conclusions? Do these prohibit publication or require revision?

Their analysis of the neutron scattering data appears to be sound. The group leaders of the neutron scattering experiments are noted experts in the field. As presented, the data acquisition and analysis discussed in the text and supplement appears to be in line with reasonable practices that any good group would use. I believe their measurements and the analysis of the data collected to be reliable.

As my expertise is in the experimental aspects of this paper, I will leave it to other reviewers to comment on the details of the theory, in particular the derivation of the effective low energy Hamiltonian.

5) Is the methodology sound? Does the work meet the expected standards in your field?

Their experimental methodology is sound. They have samples from a leading crystal growth group. The samples were mounted carefully in the relevant scattering plane with care taken to minimize the effects of indium absorption. They systematically made measurements with the CNCS spectrometer at SNS using an incident energy sufficient to cover the entire bandwidth of the low energy magnetic excitations and an incident energy giving sufficient resolution to observe their predicted sharp low energy mode. The analysis of the data appears to be sound as discussed above. They made measurements on a triple-axis instrument to demonstrate that this would likely not be observable on such an instrument.

I will leave comments on the methodology of the derivation of the effective low energy Hamiltonian to other referees. However, I would like the authors to comment on their calculated electronic structure. How does it agree with the state of the knowledge about the electronic structure in CeIn₃? I appreciate that experimental measurements of the band structure through photoemission or quantum oscillations for example can be incomplete, difficult to interpret, or not representative of bulk behavior. But given that this is a key input to their predictions, I would like the authors to comment on this point.

Along the same lines, can the authors please comment on how their quantitative predictions would change with small changes to the input parameters? I would like to exclude the possibility that the theory is "accidentally" quantitative. For instance, how much would one need to change the input parameters to quantitatively predict your experimental value $\eta_{\text{EXP}} = 2.25(4)$? If a small change to one of your inputs (for instance, a small discrepancy in the actual electronic structure not quite captured by your calculations) brings $\eta_{\text{MO-PAM}}$ into agreement with η_{EXP} , then this would tend to rule out my concern. However, if such a small change leads to a large change in $\eta_{\text{MO-PAM}}$, then this would be problematic. I emphasize that the authors don't need to comment on that exact issue specifically. The fact that they observe the predicted soft mode likely means that the quantitative predictions are robust. But I would like to hear their assessment on the quantitative limitations of their predictions.

I will leave it to other reviewers to comment on the more technical theory aspects.

6) Is there enough detail?

There is a considerable amount experimental detail. However, much of this detail comes in the Supplementary Information. In a perfect world, some of this would be in the main text, (figures S9-12, for example). However, I acknowledge that it would increase the length of the manuscript significantly and probably detract from their theory discussion to include this in the main text. With the caveat that to understand the experimental aspects, you must carefully read the supplement,

there is plenty of detail here. I leave it to the editor to decide on whether the discussion in the main text is sufficient.

I admit that I have a difficult time following the some of the details of how they derive their effective low energy theory. This is not my area of expertise and I will leave it to other referees to comment on whether there is sufficient detail presented.

---Specific comments on manuscript:

*Authors use the word "phenotypical" twice, on pages 3 and 4 of the double spaced review copy I have. It is my understanding (supported by a quick Google search and look through my college dictionary) that the English noun "phenotype" means the observable characteristics of an organism, including characteristics that come from interacting with the outside environment. "Phenotypical" is that word used as an adjective. It is not my habit to question an individual word choice of an author. That said, I strongly encourage the authors to change the language here, since CeIn₃ is not an organism and the present study is about the fundamental interactions of electrons within the material, rather than interactions with the outside world.

*The clarity of the neutron scattering data and interpretations in the Supplemental Information would be improved if the reciprocal space trajectories Γ to R, X to R, and M to R were illustrated with a simple schematic. I would recommend figure S6 as being the appropriate place, but I would be happy to have a simple map of the trajectories anywhere to make the neutron scattering slices and cuts a bit simpler to digest for someone who hasn't intimately lived with the data. In my opinion, it would be great to have such a map in the main text, but I appreciate that there may not be enough space for it. At a minimum, please add this to the supplement.

*Please improve the clarity of the caption for Figure S3 a and b. I acknowledge that I have spent some time in this review admitting my ignorance of some of the more technical theory. However, these figures have many things in them (open and closed circles, short red arrows, short black arrows, longer black arrows, numbers one through four, etc.) that simply are not discussed.

Reviewer #3:

Remarks to the Author:

In this manuscript, the authors demonstrated that a multi-orbital periodic Anderson model can be reduced to a simple Kondo-Heisenberg model for the prototypical strongly-correlated antiferromagnet CeIn₃. In this process, the complexity of the multi-orbital periodic Anderson model is greatly reduced: only 3 orbitals are kept in the effective Kondo-Heisenberg Hamiltonian out of 25 orbitals in the periodic Anderson model. More importantly, this simple Kondo-Heisenberg model is able to reproduce quantitatively the spin-wave dispersion measured by the authors using high-resolution neutron spectroscopy at ~ 1 meV level which is 10,000 times smaller than the energy scales of the electronic band structure (i.e., hopping parameters). Given the importance of heavy fermion materials and the difficulty to understand these materials quantitatively, the method proposed by authors, with hopping parameters from density functional calculations and only one fitting parameter of the Γ_7 energy level, can describe quantitatively the low-energy magnetic interactions in the prototypical Kondo lattice CeIn₃ and is very promising. It is very interesting to use the Kondo-Heisenberg model to study the superconductivity in heavy fermion materials. I support the publication of this paper in Nature Communications after the authors consider the following points:

1. Can the fitting parameter, i.e., the Γ_7 energy level be determined from ab initio calculations? It is important to derive this fitting parameter in order to make this method to have predictive power.
2. Besides the low-energy spin excitations, what other properties can this model compute?
3. Some statements in the manuscript are exaggerated. For example, the last sentence "Finally, beyond a quantitative understanding of the ground states of strongly correlated electron systems, our study paves the way for ab initio modeling of all quantum systems described by Kondo lattices." This study does not give a quantitative understanding of the ground states of strongly

correlated electron systems except for the low-energy spin excitation, nor does it have an ab initio modeling of Kondo lattices since the Γ_7 energy level remains a fitting parameter. In addition, the authors did not prove that their method is applicable to ALL Kondo materials. I suggest the authors tune down such statements.

Replies to the reviewer comments on the manuscript

Report of the First Referee #1:

First referee: *This paper reports a combined theoretical and experimental study on CeIn₃, the parent compound of Ce-based heavy fermion superconductors including CeCoIn₅. CeIn₃ itself is also a heavy fermion superconductor under pressure. Overall, the paper is well-written, and a substantial amount of work has been put into the paper by the authors. The key conclusion of the work is that a microscopic Kondo lattice model starting from ab initio electronic band structure calculations can quantitatively account for the magnetic order and excitations. Although these results are somewhat interesting, I don't see that there will be tremendous excitement about these results as expected for a Nature Communication publication. I reached this conclusion for the following reasons.*

We reply: We thank the first reviewer for appreciating our efforts and are glad to hear that they find the main conclusion of our study interesting. In the following, we provide our replies to her/his comments and suggestion.

First referee: *1. The advancements in the experimental part of the work are rather limited. I was not aware of Ref. [39]. But after carefully studying the neutron scattering work of this paper, and comparing that with Ref. [39], there is really not that much new in the present paper. Sure, the modern measurements have higher resolution, and thus are able to resolve spin excitations near the R point. But the overall spin excitations are consistent with the ref. [39].*

2. The authors argue that modern calculations can quantitatively predict magnetic exchange interactions and spin excitations. If this is conclusively demonstrated, then the paper would be quite interesting. Given that the experimental results are not in absolute units, only the dispersion part of the spectral is used to compare with the calculation.

3. Since modern neutron scattering experiments should be able to provide absolute magnetic scattering intensity by using a vanadium standard normalization, checking magnetic scattering with acoustic phonon modes, and properly correcting for the large In neutron scattering absorption, a careful neutron scattering experiment on the system should in principle be able to determine the magnetic scattering cross section in absolute units.

4. As the authors argue that their calculation can quantitatively account for the spin excitations of CeIn₃ using a microscopic Kondo lattice model from ab initio electronic bandstructure calculations, I would assume that they can also do this in absolute units. If these calculations are in absolute units, to compare with neutron scattering data in absolute units, I would then trust the results much more, and be willing to change my mind about the recommendation.

5. The work is quite interesting, but the field of heavy fermion magnetism is rather small without too many workers. I would assume that citations for this work will also be limited compared with other active fields.

6. The authors find in the concluding paragraph that both short- and long-range interactions are important for understanding magnetism in heavy Fermion materials. But this is similar to saying both itinerant electrons and localized moments are important for magnetism in many other correlated electron materials, including iron-based superconductors. Such a general conclusion does not help in solving specific problems of understanding spin excitations in heavy fermion superconductors. For example, CeCoIn₅ superconductor has a strong spin collective excitation coupled with superconductivity (resonance), but there is no understanding on the microscopic origin of the

resonance. Since CeIn5 is also a superconductor under pressure, if the authors' theory can make any qualitative or quantitative statement about the origin of superconductivity, which band is important and why magnetism might be central to the origin, the paper would find a much bigger audience and be appreciated by a

much broader community. As written, the paper is only interesting for people working on CeIn5 family of heavy fermions. Since the community is very limited, the paper will have a limited impact on the community.

We reply: Throughout her/his comments, the discussion of referee 1 touches on a few main topics repeatedly in several questions/comments. In turn, we have summarized these main topics here, and answer to each topic individually. These topics are:

(1) Referee 1 is of the opinion that our experiments while consistent with the previous work by Knafo *et al.* [Ref. 39], do not reveal new information beyond providing better resolution and therefore only provide limited new understanding.

As we describe below, we disagree with the statement of the referee that our experiments do not provide any new insights. In contrast, the vastly improved resolution, plus the choice of method (time-of-flight vs. triple-axis spectroscopy) reveal major new insights into the microscopic origins of magnetism in heavy fermion materials and make a detailed comparison with state-of-the-art theory possible. The previous work described in Ref. 39 would have been entirely unsuitable for this comparison.

Naturally, we agree with the referee that when considering the limitations with regard to resolution as well as with inherent limitations arising due to the choice of instrument, the previous measurements described in Ref. 39 are consistent with our results. However, as pointed out by the second referee, we theoretically predicted and experimentally observed “a significant deviation from the accepted knowledge of the magnetic excitations in CeIn₃”. Even assuming that the basic assumption of Ref. 39, notably that the magnetic interactions in CeIn₃ can be simply described by short-range magnetic exchange, would be correct, our measurements unambiguously demonstrate that the spin waves in CeIn₃ are not gaped. This is in stark contrast to Ref. 39, which reports a gap of larger than 1 meV compared to an overall bandwidth of about 2.5 meV! Such a substantial gap must result in a large magnetic anisotropy, which would not only be highly unconventional for a cubic material, but also is not observed in any other observable in the ground state of CeIn₃. In turn, even in this overly simplistic picture of the magnetic excitations, our result resolves a major issue with the previous measurement.

Moreover, when the part of the spin wave dispersions near the zone center, which was entirely missed (as the spin waves were thought to be gapped) by the previous measurements, is inspected, the steep spin wave dispersion immediately reveals that the simple picture of short-range interaction is insufficient because short-range exchange cannot produce this type of dispersion. More importantly, for the class of heavy fermion systems, in which magnetic order is widely believed to be mediated by the long-range RKKY interaction, it is expected that the dispersion near the zone center is steep. Despite this belief, prior to our experiments this was never observed conclusively. Naturally, this is due the inherent limitations of previously employed triple-axis measurements, which are prone to Currat-Axe spurions as we carefully demonstrate by comparing our high-resolution time-of-flight neutron spectroscopy results to a modern high-resolution triple-axis spectrometer. In turn, only the choice of a high-resolution time-of-flight spectrometer with much improved resolutions allowed for the comparison with state-of-the-art electronic structure

calculation and was an absolutely required step in understanding the complete extent of magnetic interactions in heavy fermion materials.

To conclude, our measurements, for the first time, conclusively demonstrate that long-range RKKY interactions, in addition to previously observed short-range interactions and entirely overlooked particle-particle interactions play a role in heavy fermion materials. Moreover, our theory effort can reproduce the delicate interplay of all these interactions quantitatively, which is a major step in the understanding of this class of materials.

(2) Referee 1 claims that quantitative comparison between measured spin wave excitations and theoretical calculations is only possible when neutron scattering intensity is presented in absolute units. She/he further states that calculations on absolute scale are more trustworthy, and that presenting measurements on an absolute scale would allow to change the recommendations towards supporting publication.

We agree with the referee that the intensity of the observed magnetic scattering in our work is only presented in arbitrary units and not in absolute units of μ_B^2/meV . For the improved version of the manuscript, which we submit together with this reply, we have therefore converted all intensities to absolute scale. Even though, we are naturally familiar with this conversion and have used it in some of our previous work, there are good reasons, why we did not implement absolute units in the original version of the manuscript. We provide these reasons and show how we have overcome associated challenges with the conversion in detail below. We additionally note that while performing the absorption correction only provides limited additional information, it is very time-consuming for the present case (see below for details), which was the key reason why we did not initially perform this normalization. Overcoming the challenges described below required developing new code for absorption correction due to the specific shape of the sample and constraints of our measurements. It is exactly due to this additional step that it took so long to reply to the request of the referee and produce a new version of the manuscript.

For completeness, we, however, also note that the referee's comment that quantitative comparison between measured spin wave excitations and theoretical calculations is only possible when neutron scattering intensity is presented in absolute units is incorrect. Notably, the absolute size of magnetic interactions is already encoded in the spin wave dispersion and does not require the magnetic intensity on absolute scale. Thus, the comparison of the spin wave dispersion as computed via our *ab-initio* model with the experimentally determined spin wave dispersion is indeed already quantitative, even in the absence of absolute units for the magnetic intensity and demonstrates that such modern calculations can quantitatively predict the strength of magnetic interactions.

To highlight the above point, and show that this statement is supported by the available literature, we list a number of recent examples published in leading journals where quantitative models for magnetic interactions or chemical bonds have been validated via neutron scattering measurements:

- Bo Yuan, *et al.*, Phys. Rev. X 10, 011062 (2020)
- Xiaojian Bai, *et al.*, Nature Physics 17, 467–472 (2021)
- A Scheie, *et al.*, Nature Physics 17, 726–730 (2021)
- Zheng He, *et al.*, Phys. Rev. Lett. 127, 147205 (2021)
- T. Lanigan-Atkins, *et al.*, Nature Materials 20, 977–983 (2021)
- S. E. Nikitin, *et al.*, Phys. Rev. Lett. 129, 127201 (2022)

- T. Weber, et al., Science 375, 6584 (2022)

In none of these studies the neutron intensity has been provided in absolute units, exactly because this is not necessary to make quantitative comparison between a model Hamiltonian encoding the magnetic interactions and experiment. We also note that neither referee 2 nor 3, have remarked that the comparison between experiment and theory is NOT quantitative. Referee 2, notably, has made useful comments on this comparison, which allow us to showcase how robust it is. We refer to our reply to referee 2 for this aspect.

With the quantitative relationship between spin wave dispersion and magnetic interactions firmly established, we now proceed to highlight the challenges in converting the observed inelastic magnetic scattering into absolute intensities. As the referee remarks, there are three methods according to which conversion to absolute scattering can be attempted. These are comparison to i) a standard incoherent scatterer of known mass such as vanadium, ii) phonon scattering in the studied sample, or iii) incoherent scattering in the studied samples. For our experiment, options i) and ii) were not possible, and iii) involved a major hurdle. Our experiment was originally already carried out in 2018 at the CNCS instrument at ORNL. At that time the available experimental time was not sufficient to measure a standard vanadium sample. Similarly, we had originally planned to measure an empty sample holder for optimal background subtraction, but that was also not feasible with the available time budget due to issues with the proton accelerator. Originally, it was planned to perform measurements of the vanadium standard and empty sample holder at a later stage, for example, in between two other user experiments. However, due to the heavy oversubscription of the CNCS instrument (approximately 4, which means only every fourth proposal is successful) this was not possible, especially as in the last few years, the Spallation Neutron Source had several periods during which it was not running. After this reply, we again attempted to obtain beamtime at CNCS, via collaboration with the team of scientist at ORNL, which took some time. However, it turned out to be not possible in the first place. Notably, in the meantime, several parts of the instrument have been upgraded (for example a new radial collimator was installed) and vanadium and background measurements experiment under similar conditions is not possible anymore. This excluded option i). Option ii) is not feasible because in the measured portion of momentum-energy space no suitable phonon mode is available.

This, in turn, makes option iii) the only feasible path to convert to absolute units. However, due to the strong neutron absorption of CeIn_3 arising from the large absorption cross-section of indium, and the fact that we were not able to obtain an empty sample holder measurement under identical conditions (see above) endows this method with a substantial systematic error. We note that even in the ideal case of non-absorbing samples, the systematic error of the conversion is 20% (For example, Xu et al., Rev Sci Instrum 84, 083906 (2013)). Notably, in our case, this error was much larger, because of not being able to subtract the scattering of the incoherent scattering of the sample aluminum holder due to the missing empty holder measurement. This means that to use the incoherent scattering to scale the magnetic intensity, the incoherent scattering coming from both the aluminum sample holder, as well as the sample itself, have to be considered. This introduces to additional systematic errors:

- a) The precise magnitude of the incoherent scattering from the aluminum sample holder is not exactly known because the amount of aluminum seen by the beam is not known. Careful considerations show that when the full amount of Al in the holder is used, we may overestimate the incoherent coming from the Al by up to 70%.
- b) When considering the sum of both incoherent signals, sample and holder, the absorption due to the sample cannot be considered.

Both effects result in systematic underestimation of the incoherent signal. This then results in an additional systematic error of the conversion into absolute units. It is precisely due to this large systematic error of the order of -45 and + 20 % on the resulting intensity, that we have originally refrained from plotting the magnetic intensity on an absolute scale. We note that, for the magnetic signal, an alternative route to remove the background from the sample and also to consider the effect of absorption is possible. Because, we have measurements at two different temperatures, $T = 1.8$ K within the magnetically ordered phase, and at $T = 20$ K, i.e. well above the ordered phase where no magnetic signals are present, both the background from the sample holder and all nuclear scattering can be subtracted. The resulting purely magnetic signal can then be corrected for absorption with a method that we explain in the new version of the supplementary information, which correctly takes the sample-geometry into account. It is exactly the development of this absorption correction, which took a substantial amount of time to implement and has delayed our reply. In a second step the absorption-corrected intensity is then converted into absolute units by comparison to the incoherent scattering.

Using this method all figures in the manuscript are now plotted in absolute scales and expressed as dynamical magnetic susceptibility. Similarly, assuming that the sum rule is fulfilled, and that the lifetime of the spin waves is so large that width of the observed spin waves is dominated by the experimental resolution, the theory plots have been also converted to absolute units. As an example, we show one comparison of experimental results and theory here:

We quickly summarize the main features of this comparison. First, the shape of the features in the experimental and theoretical intensity distributions are very similar. This showcases that not only does the theoretically predicted magnon dispersion fit the experimentally observed intensity maxima of the magnon peaks as we reported in the previous version of the manuscript, but that indeed the entire intensity distribution, and thus the dynamic magnetic susceptibility is matched extremely well. Second, the intensity identified in our experiment may deviate from the theory by up to 10-20%.

However, considering the above experimental systematic error, as well as accounting possible sources for systematic errors in the calculation the measured dynamic magnetic susceptibility is fully consistent with theory. We note that for entirely localized magnetic moments, the sum of the static moment of the ordered state and the fluctuating moment should be equal to the total moment expected for the relevant crystal field ground state. However, additional effects such as Kondo screening, or quantum fluctuations can reduce the moment and the resulting dynamic

magnetic susceptibility can be substantially decreased. In the related material CeRhIn₅, we found that these effects can be about 20%.

Below we show energy cuts that quantitatively compares the experimental and calculated dynamical magnetic susceptibility while including the systematic errors. This shows that experiment and theory are entirely consistent, and that our theory does not only reproduce the shape of the dispersion (and thus the magnetic interactions) quantitatively, but also reproduces the dynamical magnetic susceptibility on an absolute scale while even reproducing fine details in the structure of the intensity distribution

To conclude, we thank the referee that despite the associated challenges he/she has insisted on plotting intensities on an absolute scale, based on which we significantly improved the presentation of our result as it demonstrates the ability of our theory in even more detail. We also want to highlight that upon request of referee 2, we have compared our electronic band structure calculation, which is the basis to obtain our quantitative model with ARPES measurements. This comparison illustrates that our theory efforts are quantitatively in agreement with both charge and spin degrees of freedom in CeIn₃, showcasing the overall quantitative nature of our theory.

Finally, we note that in order to avoid possible confusions we rewrote Sec. II B of the Supplementary Material to address the relationship between the scattering cross section, $d^2\sigma/d\Omega d\omega$, and the imaginary part of the magnetic susceptibility. In addition, we recommended for further reading some textbooks for non-readers, who are no experts in the field.

(3) Referee 1 believes that our statement that both short- and long-range magnetic interactions being relevant to Kondo lattice systems is too generic and will not help to understand specific phenomena such as unconventional superconductivity.

We note that the statement that “*both short- and long-range magnetic interactions being relevant to Kondo lattice systems is too general*” directly contradicts the earlier complaint of referee 1 that our work reproduces the work of ref. 39 with higher resolution. Ref. 39, could only resolve short range interactions. In fact, while the community generally believes the long-range interactions to be present in heavy fermion systems, almost all inelastic neutron scattering measurements have been interpreted in terms of purely short-range interactions. The ability to simultaneously resolve both short and long-range interactions in a heavy fermion material may indeed provide additional

insight into how spin fluctuations influence electronic behavior including superconductivity and quantum criticality.

We further highlight that the additional claim that the statement that both short- and long-range magnetic interactions being relevant to Kondo lattice systems is “*similar to saying both itinerant electrons and localized moments are important for magnetism in many other correlated electron materials.*” is misleading as both localized and itinerant electrons can produce short- and long-range interactions and thus, there is no correspondence between these statements. For example, in insulators that have entirely localized electrons (and by correspondence only localized magnetic moments) both short- and long-range magnetic interactions may exist. An interesting case in point with technological relevance for spintronics is the material YIG in which both short-range exchange and long-range dipolar interaction play an important role in understanding the spin waves (for example, A. Serga, *et al.*, Journal of Physics D: Applied Physics, 43, 264002 (2010)). Another example is the ferromagnetic insulator EuS, in which neutron scattering could demonstrate the importance of short-range ferromagnetic exchange and long-range dipolar interactions (for example: P. Böni, M. Hennion, and J. L. Martinez, Phys. Rev. B 52, 10142(1995)). Similarly for itinerant magnets, such as elemental iron, in which the magnetic moments arise entirely due to itinerant electrons, both short-range exchange and long-range dipolar interactions are present (see S. Säubert *et al.*, Physical Review B 99, 184423 (2019)). In conclusion, a combination of localized and itinerant electrons is **not** required to have a mixture of short- and long-range magnetic interactions play an important role for materials.

In contrast, in Kondo lattice materials where both localized and itinerant electrons play a role, additional magnetic interactions emerge. These are the well-known Kondo and RKKY interactions. For the situation studied in our work, namely a material far away from quantum criticality, mostly the RKKY interaction will play a role. As explained in our manuscript, while it is widely accepted that the RKKY is the main driver for magnetic order in Kondo lattice materials, its existence has never been shown experimentally in these materials before our study. This is notably due to the very long-range nature of the RKKY interaction, which requires a very high momentum resolution, to identify the relevant part of low momentum part of the dispersion. In fact, in all previous studies of spin waves in Kondo lattice materials, the measured spin waves have been fitted with only short-range exchange type interactions, even though the accepted theory on Kondo lattice materials does **not** include any short-range exchange interactions. In addition, the low-momentum transfer signature of long-range RKKY interactions was never observed before our study.

In turn, our study is a major step forward, as it resolves this conundrum by i) experimentally revealing the low momentum signature of the RKKY interaction, and ii) showing via a quantitative theory that to establish magnetic order two further types of magnetic interactions are important in CeIn_3 , namely short-range super-exchange and particle-particle interactions.

This breakthrough is significant for all Kondo lattice materials, as, in contrast to the statement by referee 1, it already explains specific magnetic states in this class of materials. Notably, the presence of short-range interaction provides a natural explanation, why commensurate magnetic order can arise in Kondo lattice materials, which is not possible based purely on RKKY interactions. Similarly, the additional *comparable* contribution from the particle-particle channel explains the rare number of ferromagnetic *f*-electron materials, compared to antiferromagnetic cases.

Finally, concerning specifically the case of explaining unconventional superconductivity in heavy fermion materials, we agree that our theory has not yet demonstrated its full power by providing detailed understanding of this important ground state (in addition to the breakthrough of

quantitatively calculating the magnetic state at zero pressure). However, we would like to emphasize that our quantitative theory that is based on *ab-initio* band structure calculations of a real material for the first time has the potential to achieve this for any heavy fermion superconductor. This is because our microscopic and quantitative theory goes well beyond typical toy models of the Kondo lattice. This potential is also recognized by referee 3 who writes “*It is very interesting to use the Kondo-Heisenberg model to study the superconductivity in heavy fermion materials.*”

Here the effect that pressure has on the electronic structure can be included in our theory via changing the distance between cerium atoms, which will change the overlap of the *f*-electron orbitals. This will change the balance of the various involved magnetic interactions, but in particular, the interplay between the RKKY and Kondo interaction. Because our theory connects the electronic band structure with the resulting magnetic interactions and excitations, it, in turn, has the potential to provide microscopic understanding of the relationship between the superconducting gap arising at the Fermi surface and the spin resonance observed in the spin channel. Similarly, the effects of magnetic field can be included in Hamiltonian. Naturally, doing these computationally very intensive calculations as a function of lattice parameter (corresponding to hydrostatic pressure) is well beyond this manuscript, but is something that we are already actively working on.

(4) Referee 1 is concerned that our results are only of interest to a small community.

We are surprised by this statement and remind her/him that Kondo lattice models are of key importance in a growing list of quantum systems of topical interest and often with potential for applications spanning several areas of solid-state physics as described in Ref. 12-27 in our original manuscript. For completeness, we list the relevant topics with references below, and have also added further examples:

- Electronic transport through quantum dots [Ref. 12],
- Voltage-tunable magnetic moments in graphene [Ref. 13],
- Magnetism in twisted-bilayer graphene [Ref. 14] and two-dimensional organometallic materials [Ref. 15],
- The electronic structure in layered narrow-electronic-band materials [Ref. 16],
- Electronic resonances of Kagome metals [Ref. 17],
- Metallic spin liquid states [Refs. 18–20] that may even be of chiral character [Ref. 21],
- Skyrmions in centrosymmetric magnets [Refs. 22, 23].
- In hetero-structures of 2D materials composed of alternating layers of a superconductor and a Mott insulator, the coupling of both is described via the Kondo interaction and has been shown to create novel vortex phases [Eylon Persky, *et al.*, Nature volume 607, pages 692-696 (2022), now added to references] and is predicted to show quantum spin liquid physics [Shi-Zeng Lin, arXiv:2210.06550]

Kondo lattice models are also used quite widely to make progress on the theoretical understanding of a complex materials:

- Following the pioneering work by Song and Bernevig [Physical Review Letters 129, 047601(2022), also now added to the manuscript], these models have had impact on understanding flat-band materials such as twisted-bilayer graphene and others [Ref. 24] and to artificially reproduce bilayer graphene physics in

- Novel topological states such as topological superconductivity [Ref. 25] and quantum spin liquid states [Matthias Vojta, Phys. Rev. B 78, 125109 (2008) and Ref. 26], including the highly sought-after fractional quasiparticles [Ref. 27]

Just recently and after submission of our original manuscript, an additional manuscript studying a synthetic Kondo lattice in AB-stacked MoTe₂/WSe₂ moiré bilayers, in which the MoTe₂ layer is tuned to a Mott insulating state, supporting a triangular moiré lattice of local moments, and the WSe₂ layer is doped with itinerant conduction carriers was reported in Nature (Zhao, W., Shen, B., Tao, Z. et al. Gate-tunable heavy fermions in a moiré Kondo lattice. Nature 616, 61–65 (2023). <https://doi.org/10.1038/s41586-023-05800-7>). This brand-new study shows how to tune the heavy fermion quasi-particles, which are the crucial ingredient in all the quantum system listed above, and which may be exploited for voltage gates in devices. This work by Zhao *et al.*, which we now also cite in the version of the manuscript, showcases nicely how Kondo lattice physics remains of high interest to the solid-state physics community as a whole and not just researchers working on heavy fermion materials.

Our work establishes quantitative understanding of Kondo lattice physics in a very well-established prototypical heavy fermion material using an ab-initio calculation that is compared to state-of-the-art neutron scattering results. Selecting a heavy fermion material for this task, was a strategic choice, as this class of materials remains one of the best studied representations of a Kondo lattice model. The ability to quantitatively calculate the magnetic interactions and resulting spin excitations starting with an atomic-scale model is an important breakthrough for the class of heavy fermion materials, which will certainly improve the understanding of all their exotic phases as they are thought to be a result of the underlying Kondo lattice physics. This is already a major scientific step forward, because quantitative models of magnetic interactions in strongly correlated electron materials are very rare. This— in itself— warrants the interest of the broader community. In addition, the importance of our results clearly transcends the class of heavy fermion materials, as it will pave the way of similar calculations for all classes of materials mentioned above. As such we are convinced our work will be of broad interest to the community of researchers working in solid state physics and material science.

We would like to emphasize that referees 2 and 3 come to the same conclusions as we do, and in turn, support publication. For example, referee 2 writes “*Given the difficulty of making quantitative predictions in correlated electron materials, this work will be of significance to the community.*” and “*The effective low energy theory that made this prediction was developed starting from a high energy Hamiltonian and electronic structure considerations. Making reliable quantitative predictions in this fashion is highly unusual and will surely be of interest to a broad audience.*” Similarly, referee 3 summarizes “*Given the importance of heavy fermion materials and the difficulty to understand these materials quantitatively, the method proposed by authors, with hopping parameters from density functional calculations and only one fitting parameter of the Gamma₇ energy level, can describe quantitatively the low-energy magnetic interactions in the prototypical Kondo lattice CeIn₃ and is very promising.*”

Report of the Second Referee #2 (Remarks to the Author):

Second Referee: *Authors and Editors,*

My report is structured to discuss the six topics outlined in the Nature Communications review system, as well as a couple of specific comments about the manuscript at the end. I am happy to discuss any of these comments as necessary if something is not clear. My report is below:

1) Are the results noteworthy?

Quantitative theoretical predictions in correlated electron materials are notoriously difficult to make. Using the well-studied cubic Kondo lattice material CeIn₃ as a test, the authors start from a periodic Anderson model and reduce the relevant Hamiltonian to a more simple Kondo-Heisenberg model. Their model reproduces the known magnetic excitations reasonably well and predicts the existence of an unknown magnetic soft mode. They subsequently measured this mode with new, high resolution inelastic neutron scattering experiments. While not perfect, they find decent quantitative agreement with the predicted dispersion of this new mode. This is noteworthy, given that the Kondo lattice model, while applicable to a wide variety of systems, is typically thought of as a successful description if it yields qualitative predictions.

We reply: We thank the second referee for appreciating our efforts and the impact of our results. We are glad to hear that the second referee finds our study noteworthy and that she/he highlights the broad applicability of the Kondo lattice model to a wide variety of other systems.

Second referee: *2) Will the work be of significance to the field and related fields? How does it compare to the established literature?*

Given the difficulty of making quantitative predictions in correlated electron materials, this work will be of significance to the community. The authors' main accomplishment is that they predicted a significant deviation from the accepted knowledge of the magnetic excitations in CeIn₃, a very well-studied material. They then observed this deviation and found decent enough agreement with their theory when they looked with sufficient energy and momentum resolution on a modern time-of-flight neutron spectrometer. The effective low energy theory that made this prediction was developed starting from a high energy Hamiltonian and electronic structure considerations. Making reliable quantitative predictions in this fashion is highly unusual and will surely be of interest to a broad audience.

I note that their ultimate theory of the low energy excitations relies on the fact that CeIn₃ has relatively simple symmetries and is extremely well studied at this point. I question how effective this approach will be on systems that aren't as well studied or ones with more complexity. The authors suggest in their concluding paragraph that the ever increasing computational power available may lead to similar theories for other materials. While it's certainly true that more computing power will allow us to do more computationally, the low energy theories that could be derived with these methods are only as good as the starting inputs and the ability to simplify the Hamiltonians, both of which seem to be greatly assisted here by the very nature of CeIn₃.

Having said that, I don't want to take away from their main accomplishment. This will certainly be of interest, even if I am a bit more skeptical than the authors about how broadly applicable it might be.

I make a specific comment on the established literature in section 4 below.

We reply: We thank the second reviewer for pointing out the significance and the originality of our work. Further, we are thankful, that she/he point out that our study is highly unusual and of interest to a broad audience. We understand the skepticism about the applicability in the near future, which as the second referee notes is not diminishing our accomplishments. Nevertheless, we do find this comment important and have adjusted the respective sentence in the manuscript accordingly.

Second referee: 3) *Does work support conclusions and claims, or is additional evidence needed?*

Assuming the derivation of their low energy theory is robust, and that the authors can address the other concerns outlined here, then I think the work does generally support their conclusions. My reservations about the broad applicability of the technique presented (at least in the near or medium term) are outlined above. To overcome this limitation would require a lot more theory work and experiments on different materials. That would clearly be beyond the scope of this manuscript, so I don't think my general reservations should preclude publication.

However, in this spirit, I ask the authors to reconsider one of their sentences towards the end of the paper: "We note that the ability to resolve these steep low-energy spin excitations also ushers in high-resolution TOF spectroscopy as a novel technique to verify the electronic band structure of magnetically ordered heavy-fermion materials." As I said, I'm not as convinced as they are about how this can be applied to other systems in the immediate future.

I will leave it to other reviewers to comment on the more technical theory aspects.

We reply: We thank the second reviewer for these remarks and for recommending our manuscript for publication. We understand that she/he finds the statement exaggerated and adjusted the respective sentence as follows: "We note that the ability to resolve these steep low-energy spin excitations also ushers in high-resolution TOF spectroscopy as a complementary technique to access the electronic band structure of magnetically ordered heavy-fermion materials." We are glad to read that our study may motivate more theory work and experiments on other materials.

Second referee: 4) *Are there any flaws in data analysis, interpretation, and conclusions? Do these prohibit publication or require revision?*

Their analysis of the neutron scattering data appears to be sound. The group leaders of the neutron scattering experiments are noted experts in the field. As presented, the data acquisition and analysis discussed in the text and supplement appears to be in line with reasonable practices that any good group would use. I believe their measurements and the analysis of the data collected to be reliable.

As my expertise is in the experimental aspects of this paper, I will leave it to other reviewers to comment on the details of the theory, in particular the derivation of the effective low energy Hamiltonian.

We reply: We are thankful for this assessment and are happy to read that the presentation of data appears to be reliable.

Second referee: 5) *Is the methodology sound? Does the work meet the expected standards in your field?*

Their experimental methodology is sound. They have samples from a leading crystal growth group. The samples were mounted carefully in the relevant scattering plane with care taken to minimize the effects of indium absorption. They systematically made measurements with the CNCS spectrometer at

SNS using an incident energy sufficient to cover the entire bandwidth of the low energy magnetic excitations and an incident energy giving sufficient resolution to observe their predicted sharp low energy mode. The analysis of the data appears to be sound as discussed above. They made measurements on a triple-axis instrument to demonstrate that this would likely not be observable on such an instrument.

I will leave comments on the methodology of the derivation of the effective low energy Hamiltonian to other referees. However, I would like the authors to comment on their calculated electronic structure. How does it agree with the state of the knowledge about the electronic structure in CeIn_3 ? I appreciate that experimental measurements of the band structure through photoemission or quantum oscillations for example can be incomplete, difficult to interpret, or not representative of bulk behavior. But given that this is a key input to their predictions, I would like the authors to comment on this point.

We reply: We are glad to read that our methodological approach is sound to the second referee. We agree that it is a good idea to compare our DFT calculations with previously reported experimental works. The electronic structure of CeIn_3 was investigated experimentally by photoemission in Ref.: Y. Zhang, et al. (Sci Rep , 33613 (2016)). In Fig. 2 of this reference, the authors presented the valence band structure of CeIn_3 :

As proposed by the second referee, we compared the results of our DFT calculations with these experimental data. For this quest, we extracted the photoemission intensity from subpanels c1 and d1 and compared it (represented by color-plot) with our DFT calculations (black lines), as shown in the following:

In addition, we compared our calculations (black lines) with the electron-band trajectories (orange dotted lines) that Zhang et al. inferred from their data:

In conclusion, we find the agreement between our results and the data extremely sound, although we note the challenges that arise when comparing surface sensitive ARPES data to DFT calculations and irrespective of the difficulties in the interpretation of ARPES data recorded in cubic materials. Notably, in cubic materials the momentum resolution along the surface normal can be broadened resulting in difficulties of interpreting the ARPES data [see for example: M. Rahn *et al.* Nature Comm. 13, 6129 (2022)].

In order to highlight the consistency between our band structure calculation and previous ARPES measurements, we now include the above comparison in the Supplementary Information to our manuscript.

Second referee: *Along the same lines, can the authors please comment on how their quantitative predictions would change with small changes to the input parameters? I would like to exclude the possibility that the theory is “accidentally” quantitative. For instance, how much would one need to change the input parameters to quantitatively predict your experimental value $\eta_{EXP} = 2.25(4)$? If a small change to one of your inputs (for instance, a small discrepancy in the actual electronic structure not quite captured by your calculations) brings η_{MO-PAM} into agreement with η_{EXP} , then this would tend to rule out my concern. However, if such a small change leads to a large change in η_{MO-PAM} , then this would be problematic. I emphasize that the authors don’t need to comment on that exact issue specifically. The fact that they observe the predicted soft mode likely means that the quantitative predictions are robust. But I would like to hear their assessment on the quantitative limitations of their predictions.*

I will leave it to other reviewers to comment on the more technical theory aspects.

We reply: We thank the referee for bringing up this important point. To assess the robustness of the theoretical results, we have computed the interaction and the magnon dispersion for a new value of the chemical potential $E_F=12.598$ eV ($E_F=12.588$ eV in the manuscript), which is within the range of accuracy (~ 10 meV) of the DFT calculation. In the following figure, we show the comparison between the magnon dispersion obtained for both values of the Fermi level. The red lines correspond to $E_F=12.588$ eV, and dashed lines to the new value $E_F=12.598$ eV. As it is clear from the figure, the magnon dispersions are practically the same and the relative change in the value of η_{MO-PAM} is $\sim 1\%$

Second referee: 6) Is there enough detail?

There is a considerable amount experimental detail. However, much of this detail comes in the Supplementary Information. In a perfect world, some of this would be in the main text, (figures S9-12, for example). However, I acknowledge that it would increase the length of the manuscript significantly and probably detract from their theory discussion to include this in the main text. With the caveat that to understand the experimental aspects, you much carefully read the supplement, there is plenty of detail here. I leave it to the editor to decide on whether the discussion in the main text is sufficient.

We reply: We thank the second reviewer for pointing out that all details of the experimental study are provided either in the main text or the Supplementary Material and agree that the information selected in the manuscript had to be deliberately selected from the large amount of experimental data. We are glad to hear that also the material that we presented in the Supplementary Material has a profound quality that would warrant publication in the manuscript.

Second referee: *I admit that I have a difficult time following the some of the details of how they derive their effective low energy theory. This is not my area of expertise and I will leave it to other referees to comment on whether there is sufficient detail presented.*

---Specific comments on manuscript:

**Authors use the word “phenotypical” twice, on pages 3 and 4 of the double spaced review copy I have. It is my understanding (supported by a quick Google search and look through my college dictionary) that the English noun “phenotype” means the observable characteristics of an organism, including characteristics that come from interacting with the outside environment. “Phenotypical” is that word used as an adjective. It is not my habit to question an individual word choice of an author. That said, I strongly encourage the authors to change the language here, since CeIn3 is not an organism and the present study is about the fundamental interactions of electrons within the material, rather than interactions with the outside world.*

We reply: We thank the referee for pointing this out, have changed the wording in the two occurrences in the manuscript.

Second referee: **The clarity of the neutron scattering data and interpretations in the Supplemental Information would be improved if the reciprocal space trajectories Γ to R, X to R, and M to R were illustrated with a simple schematic. I would recommend figure S6 as being the appropriate place, but I would be happy to have a simple map of the trajectories anywhere to make the neutron scattering slices and cuts a bit simpler to digest for someone who hasn't intimately lived with the data. In my opinion, it would be great to have such a map in the main text, but I appreciate that there may not be enough space for it. At a minimum, please add this to the supplement.*

We reply: We thank the referee for this remark as we entirely agree that this adjustment improves the clarity of representation. For a better readability, we put this overview of high-symmetry points in momentum space already in Figure 2 of the submitted version of the main text. In addition, we adapted the color coding and labelling of reciprocal space positions in each figure of the manuscript as well as of the supplement.

Second referee: **Please improve the clarity of the caption for Figure S3 a and b. I acknowledge that I have spent some time in this review admitting my ignorance of some of the more technical theory. However, these figures have many things in them (open and closed circles, short red arrows, short black arrows, longer black arrows, numbers one through four, etc.) that simply are not discussed.*

We reply: We agree with the second referee and changed the caption of Figure S3.

Reports of the Third Referee #3 (Remarks to the Author):

Third referee: *In this manuscript, the authors demonstrated that a multi-orbital periodic Anderson model can be reduced to a simple Kondo-Heisenberg model for the prototypical strongly-correlated antiferromagnet CeIn₃. In this process, the complexity of the multi-orbital periodic Anderson model is greatly reduced: only 3 orbitals are kept in the effective Kondo-Heisenberg Hamiltonian out of 25 orbitals in the periodic Anderson model. More importantly, this simple Kondo-Heisenberg model is able to reproduce quantitatively the spin-wave dispersion measured by the authors using high-resolution neutron spectroscopy at ~1 meV level which is 10,000 times smaller than the energy scales of the electronic band structure (i.e., hopping parameters). Given the importance of heavy fermion materials and the difficulty to understand these materials quantitatively, the method proposed by authors, with hopping parameters from density functional calculations and only one fitting parameter of the Gamma₇ energy level, can describe quantitatively the low-energy magnetic interactions in the prototypical Kondo lattice CeIn₃ and is very promising. It is very interesting to use the Kondo-Heisenberg model to study the superconductivity in heavy fermion materials. I support the publication of this paper in Nature Communications after the authors consider the following points:*

We reply: We thank the third reviewer for this assessment, for pointing out that our study is very promising, and for recommending publication in Nature Communications. We addressed the points suggested by the third referee, as explained in the following.

Third referee: *1. Can the fitting parameter, i.e., the Gamma₇ energy level be determined from ab initio calculations? It is important to derive this fitting parameter in order to make this method to have predictive power.*

We reply: We thank the third referee for this question as it allows us to clarify further details of our model. The starting point of our model— the DFT calculation and the resulting tight-binding model — assume that the *f*-electrons are itinerant. However, in real heavy fermion materials the *f*-electrons are localized by a strong intra-atomic Coulomb *U* interaction. In contrast, as shown by the pioneering work of Yamada for CeB₆ and reproduced for CeIn₃ in our work, the conduction electron bands will essentially remain unchanged. The energy of the Gamma₇ doublet cannot be extracted from ab-initio calculations because, once the *f*-electrons are localized by a strong on-site repulsion *U*, the Gamma₇ energy level can lie anywhere below the Fermi level. This leaves a *single* free parameter in the Periodic Anderson model, which does not eliminate its predictive power because it can be fixed by only one experiment. The resulting model can then be used to predict other experimental results. Moreover, one free parameter is in general not enough to fit a full magnon dispersion and the intensities of the magnon peaks, which is what we are doing in the new version of the manuscript.

We note that this is the rule for most attempts of adding correlations to density functional theory. The primary example of a free parameter is the on-site Coulomb repulsion *U* included in DFT+*U* and LDA+DMFT. Since different estimates of *U* differ significantly, *U* is typically fixed by fitting one experiment and the resulting model is then used to predict other experiments. The Gamma₇ energy level is the only free parameter in our calculation because we are assuming infinite *U* (i.e. eliminating the *f*² configuration). All the other parameters of the Periodic Anderson Model are derived from first principles.

For the case under consideration (magnetically ordered Kondo lattices), the energy scale of the Gamma₇ level is set by overall bandwidth of the magnons. We exploit that for our effective

magnetic Kondo lattice model, there is a one-to-one correspondence between the Γ_7 energy level and the bandwidth of magnetic excitations. After this calibration, our model obtains quantitative agreement on energy scales of sub meV energy scales. We further highlight that even in the absence of this calibration, the shape of the dispersion would be quantitatively correct modulo this calibration factor provided by the Γ_7 . So the procedure is similar to many other material specific studies, where the Coulomb U in the band structure calculations is calibrated with experimental data. For example, even in DMFT calculations, which may consider the local intra-atomic Coulomb interaction correctly, the material dependent strength of U needs to be calibrated by comparing to experimental energy scales. For the simpler case of CePd_3 — a Kondo lattice in the limit of vanishing RKKY interactions — this was achieved via comparison to photo emission spectra [M. Rahn *et al.* Nature Comm. 13, 6129 (2022)]. We note that this is a far easier case because for vanishing RKKY interactions the f electrons are not completely localized. Nevertheless, the degree of normalization needs to be calibrated against experiments at higher energy scales to obtain a quantitative effective model at the relevant energy scales close to the Fermi level. The resulting effective model contains the correct physics on an absolute scale that allows to make quantitative predictions. We note that the value of the Γ_7 energy level then fixes the Kondo scale for our Kondo Heisenberg model. As such, this enables testable consequences for the charge degrees of freedom when the model is solved by suitable many-body techniques. This is work in progress.

We also note that upon the request of referee 1, we have now also calculated the full dynamical spin susceptibility and normalized our experimentally observed magnetic intensity on an absolute scale; this shows that our model even reproduces the neutron scattering intensities on an absolute scale illustrating the predictive power of the model. Similarly (see also below), we can calculate the magnetic ordering vector from our model.

Third referee: 2. Besides the low-energy spin excitations, what other properties can this model compute?

We reply: We are thankful for this question. As pointed out already in the manuscript, our model permits, on the one hand, to identify three different types of magnetic interactions that are crucial for the emergence of magnetism in CeIn_3 and in a broader sense potentially also for magnetism in f -electron metals in general. On the other hand, our study predicts *ab-initio* the emergence of $(\pi\pi\pi)$ antiferromagnetism with the ordering vector located at the R point. Restricting the comment to the low energy spin Hamiltonian that we have derived, we can compute any static or dynamic spin response. For instance, one can compute the spin-lattice relaxation rate from the spin excitations in our model. By including the excited Γ_8 CEF level one would also be able to compute the high field magnetization $M(H)$ and the H-T phase diagram. This is also currently work in progress. More broadly, our Kondo-Heisenberg model would enable calculations of the charge response: superconductivity, Fermi surface evolution/reconstruction, renormalization of electronic mass, Kondo temperature, transport properties, etc.

Third referee: 3. *Some statements in the manuscript are exaggerated. For example, the last sentence "Finally, beyond a quantitative understanding of the ground states of strongly correlated electron systems, our study paves the way for ab initio modeling of all quantum systems described by Kondo lattices." This study does not give a quantitative understanding of the ground states of strongly correlated electron systems except for the low-energy spin excitation, nor does it have an ab initio modeling of Kondo lattices since the Γ_7 energy level remains a fitting parameter. In addition,*

the authors did not prove that their method is applicable to ALL Kondo materials. I suggest the authors tune down such statements.

We reply: We understand that the third reviewer finds this sentence exaggerated. Therefore, we modified the statement in the manuscript.

Summary of all major changes to the manuscript and the Supplementary Material

- **(C1)** Following the suggestions of the first reviewer on our experimental data, we performed a comprehensive, sample-geometry dependent absorption correction and normalized the magnetic spectroscopy intensity to the incoherent scattering of Indium, in order to bring the recorded neutron spectroscopy intensity on an absolute scale. Having performed these changes, the presented neutron spectroscopy data now reflect the size of the imaginary part of the dynamic magnetic susceptibility on an absolute scale. The described changes were implemented in Figs. 3 and 4 of the main text and in Figs S7, S8, S9, S10, S11, S12, and S13 of the Supplementary Material. A detailed account on these changes is provided in Sec. II B of the Supplementary Material.
- **(C2)** We repeated all relevant fits to the corrected neutron spectroscopy data. The revised fitting parameters include the bandwidth of excitations, the velocity of magnons, and the η -parameter. The changes in the flow-text of the manuscript and the Supplementary Material are indicated in color. In addition, we replaced Table S1 of the Supplementary Material
- **(C3)** We implemented minor changes in the manuscript and indicated all of them in blue color.
- **(C4)** We noticed that a factor of 1/2 was missing in Eq. S13, S15-S17 of the Supplementary Material, which leads to a different definition of S14a-f. These modifications implied a slightly different value for the energy of the f-band (new value: $E_f = 12.005$ eV; old value: $E_f = 12.069$ eV) and an almost unnoticeable change in all calculations of excitations and interactions presented.
- **(C5)** We added section X. in the Supplementary Material, where we present a comparison of our DFT calculations with the band-structure obtained by angle-resolved photo emission spectroscopy
- **(C6)** Upon the request of the editor, we have included data and code availability statements at the end of the manuscript.

Reviewers' Comments:

Reviewer #2:

Remarks to the Author:

My original report on this manuscript featured 6 broad points and two additional specific points. The authors addressed all of these. This review of the revised manuscript is structured to give my thoughts on their response to these points and the relevant manuscript changes. In summary, I am satisfied with the revised manuscript and see no reason why it should not be published. I outline a couple of potential small changes below in points 5 and 6. I leave it to the editor to decide whether these are necessary. I also leave it to the editor, other referees, and authors to decide whether the concerns of the other referees have been satisfied. I am happy elaborate on any aspect if needed.

1) No comments.

2) The changes in response to my comments are fine.

3) The changes in response to my comments are fine.

4) No comments.

5) Thank you for including this comparison and discussing the electronic structure inputs. I appreciate that the experimental information on electronic structure was in print elsewhere, but the direct comparison you make to your DFT calculation was important, given how central it is to the main result. I am satisfied with the comparison, its inclusion in the supplement, and the associated discussion.

I also thank the authors for looking at how a small input change would affect their results. I agree that there is very little effect. Perhaps it is worth a comment in the main text, but I leave it to the editor to decide if that is necessary.

6) Thank you for clarifying this point. I must admit that I missed the reciprocal space map in figure 2 when I read the manuscript originally. It is precisely what I was asking the authors to add. On re-reading the paper, I think the reason why I missed it is that it is not discussed anywhere and so I did not give it any of my attention. I appreciate that the map is self-explanatory, but as my own experience with the manuscript has just shown, it is probably better to discuss everything that appears to avoid this sort of confusion. I would suggest a sentence be added to the caption, and/or a brief mention in the text. I leave it to the editor to decide if this is necessary.

Specific comment 1) The changes in response to my comment are fine.

Specific comment 2) The revised caption to figure S3 is much more clear. The original text describing the schematic diagrams may have been fine for someone who looks at the details of virtual processes all the time. But for someone more invested in the experimental end of condensed matter physics, the revised text is much better. The revised text is also more appropriate for a broad journal such as Nature Communications.

Reviewer #3:

Remarks to the Author:

In the revised manuscript and authors' response to previous review report, all my questions/comments have been addressed. Therefore, I recommend to publish the manuscript in Nature Communications in its current form.

Detailed Replies to Reviewer Comments.

Reviewer #2 (Remarks to the Author):

My original report on this manuscript featured 6 broad points and two additional specific points. The authors addressed all of these. This review of the revised manuscript is structured to give my thoughts on their response to these points and the relevant manuscript changes. In summary, I am satisfied with the revised manuscript and see no reason why it should not be published. I outline a couple of potential small changes below in points 5 and 6. I leave it to the editor to decide whether these are necessary. I also leave it to the editor, other referees, and authors to decide whether the concerns of the other referees have been satisfied. I am happy elaborate on any aspect if needed.

Our reply:

We thank the referee for her/his additional review and the detailed explanation on how our revised manuscript addresses her/his previous remarks and comments. We are glad to read that she/he now supports publication of the revised manuscript. Below we respond to the minor points she/he raised in this round of comments.

Specific comments of reviewer 1.

1) No comments.

2) The changes in response to my comments are fine.

Our reply:

We appreciate that the reviewer clarifies that our changes address her/his question.

3) The changes in response to my comments are fine.

Our reply:

We appreciate that the reviewer clarifies that our changes address her/his question.

4) No comments.

5) Thank you for including this comparison and discussing the electronic structure inputs. I appreciate that the experimental information on electronic structure was in print elsewhere, but the direct comparison you make to your DFT calculation was important, given how central it is to the main result. I am satisfied with the comparison, its inclusion in the supplement, and the associated discussion.

I also thank the authors for looking at how a small input change would affect their results. I agree that there is very little effect. Perhaps it is worth a comment in the main text, but I leave it to the editor to decide if that is necessary.

Our reply:

We thank the reviewer for her/his additional remarks on this point. We also feel that the direct comparison of our electronic structure calculations with previously published ARPES data in the supplement showcases the robustness of our results better. Concerning the changes of the input parameter, we have now added the following phrase to the manuscript on page 7 at the end of the 2nd paragraph:

“In addition to being quantitatively accurate, our calculations are robust against small changes of the chemical potential of the order of 10 meV (resolution of our band structure calculation), which modify the slope and bandwidth of the magnon dispersion by less than 1%.”

6) Thank you for clarifying this point. I must admit that I missed the reciprocal space map in figure 2 when I read the manuscript originally. It is precisely what I was asking the authors to add. On re-reading the paper, I think the reason why I missed it is that it is not discussed anywhere and so I did not give it any of my attention. I appreciate that the map is self-explanatory, but as my own experience with the manuscript has just shown, it is probably better to discuss everything that appears to avoid this sort of confusion. I would suggest a sentence be added to the caption, and/or a brief mention in the text. I leave it to the editor to decide if this is necessary.

Our reply:

Thanks for pointing out that the reciprocal space map is easy to overlook. We have added the following comment to the main text, right after the path in reciprocal space is mentioned for the first time on page 6 (last paragraph) of the manuscript:

“The inset on the upper right corner of Fig. 2b shows the position of these high symmetry points in the Brillouin zone that define this path.”

We have also adjusted the captions of figures 2 and 3 to point out where the map can be found.

Specific comment 1) The changes in response to my comment are fine.

Our reply:

We are glad that the reviewer agrees with the changes we made to the revised manuscript in order to accommodate her/his comments.

Specific comment 2) The revised caption to figure S3 is much more clear. The original text describing the schematic diagrams may have been fine for someone who looks at the details of virtual processes all the time. But for someone more invested in the experimental end of condensed matter physics, the revised text is much better. The revised text is also more appropriate for a broad journal such as Nature Communications.

Our reply:

The previous comments of the reviewer made about the original version of the manuscript were useful in order to make these improvements. We are glad that our adjustments improved the readability of the text for the readers of Nature Communications.

Reviewer #3 (Remarks to the Author):

In the revised manuscript and authors' response to previous review report, all my questions/comments have been addressed. Therefore, I recommend to publish the manuscript in Nature Communications in its current form.

Our reply:

We thank the reviewer for looking at the revised manuscript and her/his support for the publication in Nature Communication.